

# Impact of dust addition on the metabolism of Mediterranean plankton communities and carbon export under present and future conditions of pH and temperature

Frédéric Gazeau[1], France Van Wambeke[2], Emilio Marañón[3], María Pérez-Lorenzo[3], Samir Alliouane[1], Christian Stolpe[1], Thierry Blasco[1], Nathalie Leblond[4], Birthe Zäncker[5,6], Anja Engel[6], Barbara Marie[7], Julie Dinasquet[7,8], Cécile Guieu[1]

[1] Sorbonne Université, CNRS, Laboratoire d'Océanographie de Villefranche, LOV, 06230 Villefranche-sur-Mer, France

[2] Aix-Marseille Université, Université de Toulon, CNRS/INSU, IRD, Mediterranean Institute of Oceanography (MIO), UM 110, 13288, Marseille, France

[3] Department of Ecology and Animal Biology, Universidade de Vigo, 36310 Vigo, Spain

[4] Sorbonne Université, CNRS, Institut de la Mer de Villefranche, IMEV, 06230 Villefranche-sur-Mer, France

[5] The Marine Biological Association of the UK, PL1 2PB Plymouth, United Kingdom

[6] GEOMAR Helmholtz Centre for Ocean Research, Kiel, Germany

[7] CNRS, Sorbonne Université, Laboratoire d'Océanographie Microbienne, LOMIC, F-66650 Banyuls-sur-Mer, France

[8] Scripps Institution of Oceanography, University of California San Diego, USA

Correspondence to: Frédéric Gazeau (f.gazeau@obs-vlfr.fr)

Keywords: Mediterranean Sea; Atmospheric deposition; Plankton community metabolism; Carbon export; Ocean acidification; Ocean warming



## Abstract

Although atmospheric dust fluxes from arid as well as human-impacted areas represent a
significant source of nutrients to surface waters of the Mediterranean Sea, studies focusing on the
evolution of the metabolic balance of the plankton community following a dust deposition event
are scarce and none were conducted in the context of projected future levels of temperature and
pH. Moreover, most of the experiments took place in coastal areas. In the framework of the
PEACETIME project, three dust-addition perturbation experiments were conducted in 300-L
tanks filled with surface seawater collected in the Tyrrhenian Sea (TYR), Ionian Sea (ION) and
in the Algerian basin (FAST) onboard the R/V "Pourquoi Pas?" in late spring 2017. For each
experiment, six tanks were used to follow the evolution of chemical and biological stocks,
biological activity and particle export. The impacts of a dust deposition event simulated at their
surface were followed under present environmental conditions and under a realistic climate
change scenario for 2100 (ca. + 3 °C and -0.3 pH units). The tested waters were all typical of
stratified oligotrophic conditions encountered in the open Mediterranean Sea at this period of the
year, with low rates of primary production and a metabolic balance towards net heterotrophy.
The release of nutrients after dust seeding had very contrasting impacts on the metabolism of the
communities, depending on the station investigated. At TYR, the release of new nutrients was
followed by a negative impact on both particulate and dissolved [14]C-based production rates,
while heterotrophic bacterial production strongly increased, driving the community to an even
more heterotrophic state. At ION and FAST, the efficiency of organic matter export due to
mineral/organic aggregation processes was lower than at TYR likely related to a lower
quantity/age of dissolved organic matter present at the time of the seeding. At these stations, both



the autotrophic and heterotrophic community benefited from dust addition, with a stronger
relative increase in autotrophic processes observed at FAST. Our study showed that the potential
positive impact of dust deposition on primary production depends on the initial composition and
metabolic state of the investigated community. This potential is constrained by the quantity of
nutrients added in order to sustain both the fast response of heterotrophic prokaryotes and the
delayed one of primary producers. Finally, under future environmental conditions, heterotrophic
metabolism was overall more impacted than primary production, with the consequence that all
integrated net community production rates decreased with no detectable impact on carbon
export, therefore reducing the capacity of surface waters to sequester anthropogenic $CO_2$.



## 1. Introduction

Low Nutrient Low Chlorophyll (LNLC) areas represent 60% of the global ocean surface
area (Longhurst et al., 1995) and, although phytoplankton production there is limited by the
availability of nitrogen, phosphorus and iron, it accounts for 50% of global carbon export
(Emerson et al., 1997). Atmospheric dust fluxes from arid as well as anthropogenic sources
represent a significant source of these nutrients to surface waters in these regions and as such
could play a significant role in stimulating primary production (e.g. Bishop et al., 2002; Guieu et
al., 2014b; Jickells and Moore, 2015), potentially increasing the efficiency of the biological
pump in the sequestration of atmospheric $CO_2$. However, as heterotrophic prokaryotes have been
shown to outcompete phytoplankton during nutrient addition experiments (e.g. Guieu et al.,
2014a; Mills et al., 2008; Thingstad et al., 2005), dust deposition could induce even stronger
enhancements of heterotrophic bacterial production and/or respiration rates thereby reducing net
atmospheric $CO_2$ drawdown and the potential for carbon export outside the euphotic zone (Guieu
et al., 2014b). Indeed, several experiments conducted in the Atlantic Ocean and in the
Mediterranean Sea have shown a fast and dominant effect of dust additions on heterotrophic
bacterioplankton metabolism (Herut et al., 2005, 2016; Lekunberri et al., 2010; Marañón et al.,
2010; Pulido-Villena et al., 2008, 2014). However, to the best of our knowledge, no study
focused on the evolution of the metabolic balance of the plankton community after such a dust
event in the open sea. The metabolic balance (or net community production, NCP) is defined as
the difference between gross primary production (GPP) of autotrophic organisms and community



respiration (CR) of both autotrophic and heterotrophic organisms, revealing the capacity of a
system to sequester carbon via the biological pump.

The Mediterranean Sea is a perfect example of LNLC regions and receives anthropogenic

aerosols originating from industrial and domestic activities from all around the basin and other
parts of Europe and pulses of natural inputs from the Sahara (e.g. Bergametti et al., 1989;
Desboeufs et al., 2018). These atmospheric depositions, mostly in the form of pulsed inputs
(Loÿe-Pilot and Martin, 1996), provide new nutrients (Guieu et al., 2010; Kouvarakis et al.,
2001; Markaki et al., 2003; Ridame and Guieu, 2002) to the surface waters with fluxes that are of
the same order of magnitude as riverine inputs (Powley et al., 2017). These significant nutrient
enrichments likely support primary production especially during the stratification period (Bonnet
et al., 2005; Ridame and Guieu, 2002), however no clear correlation between dust and ocean
color have been evidenced from long series of satellite observations (Guieu and Ridame, 2020).
This raises the question on which compartment (autotrophic or heterotrophic) benefits the most
from these transient relieves in nutrient limitation.

In response to ocean warming and increased stratification, LNLC areas are expected to

expand in the future (Irwin and Oliver, 2009; Polovina et al., 2008) due to lower nutrient supply
from sub-surface waters (Behrenfeld et al., 2006). Furthermore, dust deposition could increase in
the future due to desertification (Moulin and Chiapello, 2006), although so far the trend for
deposition remains uncertain because the drying of the Mediterranean basin might also induce
less wet deposition over the basin (Laurent et al., 2021). Nevertheless, whether the fluxes
increase or not in the coming decades and centuries, new nutrients from atmospheric sources will
play an important role in a surface mixed layer even more stratified and isolated from the deeper
nutrient-rich layer. The question remains on how plankton metabolism and carbon export would





respond in a warmer and more acidified ocean. Indeed, with an average annual anthropogenic
$CO_2$ uptake, during the period 2010 to 2019, of $2.5 \pm 0.6$ GtC (~22.9% of anthropogenic
emissions; Friedlingstein et al., 2020), the oceans substantially contribute towards slowing down
the increase in atmospheric $CO_2$ concentrations, and therefore towards limiting terrestrial and
ocean warming. However, this massive $CO_2$ input induces global changes in seawater chemistry
referred to as "ocean acidification" because increased $CO_2$ concentration lowers seawater pH
(i.e. increases its acidity).

Although the response of plankton metabolism to ocean warming has been shown to be

highly dependent on resource availability (Lewandowska et al., 2014), both for heterotrophic
bacteria (Lopez-Urrutia and Moran, 2007) and phytoplankton (Marañón et al., 2018), it has been
suggested that ocean warming will substantially weaken the ocean $CO_2$ sink in the future as a
consequence of stronger increase in remineralization than in photosynthesis processes, following
the metabolic theory of ecology (MTE; Brown et al., 2004; Gillooly et al., 2001). Ocean
acidification alone has been shown to exert no or very limited influence on plankton metabolism
in the Mediterranean Sea (Maugendre et al., 2017a; Mercado et al., 2014). To the best of our
knowledge, only Maugendre et al. (2015) studied the combined impact of ocean warming and
acidification on plankton metabolism in the Mediterranean Sea. They found a very limited
impact of ocean acidification on the plankton community and a positive impact of warming on
small phytoplankton species (e.g. Cyanobacteria) with a potential decrease of the export and
energy transfer to higher trophic levels. Nevertheless, that study was conducted under nutrient
depleted conditions and there is still a need to assess the combined impact of warming and
acidification on the metabolic balance of plankton communities in this region, following a
transient relief in nutrient availability (Maugendre et al., 2017b).



121  So far there has been no attempt to evaluate the evolution of plankton metabolism and

122 carbon export following atmospheric deposition in the context of future levels of temperature

123 and pH. Such experiments were conducted in the frame of the PEACETIME project (ProcEss

124 studies at the Air-sEa Interface after dust deposition in the MEditerranean sea; http://peacetime-

125 project.org/) during the cruise on board the R/V "Pourquoi Pas?" in May/June 2017 (Guieu et al.,

126 2020). The project aimed at extensively studying and parameterizing the chain of processes

127 occurring in the Mediterranean Sea after atmospheric deposition, especially of Saharan dust, and

128 to put them in perspective of on-going environmental changes. During this cruise, three

129 perturbation experiments were conducted in 300-L tanks filled with surface seawater collected in

130 the Tyrrhenian Sea (TYR), Ionian Sea (ION) and in the Algerian basin (FAST; Fig. 1). Six tanks

131 were used to follow the evolution of chemical and biological stocks, biological activity and

132 export, following a wet dust deposition event simulated at their surface, both under present

133 environmental conditions and following a realistic climate change scenario for 2100 (ca. + 3 °C

134 and -0.3 pH units; IPCC, 2013). A companion paper presents the general setup of the

135 experiments and the impacts of dust under present and future environmental conditions on

136 nutrients and biological stocks (Gazeau et al., 2020). Here, we focus on the impacts of dust

137 seeding on plankton metabolism (e.g. primary production, heterotrophic prokaryote production)

138 and carbon export.




## 2. Material and Methods

## 2.1. General set-up

The general set-up of the experiments is fully detailed in Gazeau et al. (2020). Briefly,
three experiments were performed at the long duration stations TYR, ION and FAST during the
Peacetime cruise onboard R/V "Le Pourquoi Pas?" (Fig. 1). During these experiments (3 to 4
days each), seawater was incubated in 300-L tanks (Fig. S1) installed in a temperature-controlled
container, in which the irradiance spectrum and intensity can be finely controlled and in which
future ocean acidification and warming conditions can be fully reproduced. The tanks were made
of high-density polyethylene (HDPE) and were trace-metal free in order to avoid contaminations,
with a height of 1.09 m, a diameter of 0.68 m, a surface area of 0.36 $m^2$ and a volume of 0.28 $m^3$.
The conical base of the tanks was equipped with a sediment trap that was left open during the
duration of the experiments and removed at the end. The experimental protocol comprised two
unmodified control tanks (C1 and C2), two tanks enriched with Saharan dust (D1 and D2) and
two tanks enriched with Saharan dust and maintained simultaneously under warmer (+ 3 ºC) and
acidified (-0.3 pH unit) conditions (G1 and G2). At the three stations, tanks were always filled at
the end of the day before the start of the experiments: TYR (17/05/2017), ION (25/05/2017) and
FAST (02/06/2017). The tanks were filled by means of a large peristaltic pump (Verder© VF40
with EPDM hose, flow of 1200 L $h^{-1}$) collecting seawater below the base of the boat (depth of ~
5 m), used to supply continuously surface seawater to a series of instruments during the entire
campaign. While filling the tanks, seawater was sampled for the measurements of selected
parameters (sampling time = t-12h). After filling the tanks, seawater was slowly warmed





overnight using 500 W heaters, controlled by temperature-regulation units (COREMA©), in G1
and G2 to reach an offset of + 3 °C. $^{13}$C-bicarbonate was added to all tanks at 4:00 am (all times
in local time) and G1 and G2 were acidified by addition of $CO_2$-saturated filtered (0.2 μm)
seawater (~1.5 L in 300 L; collected when filling the tanks at each station) at 4:30 am to reach a
pH offset of -0.3. Sampling for many parameters took place prior to dust seeding (sampling time
= t0). Dust seeding was performed between 7:00 and 9:00 in tanks D1, D2, G1 and G2. The same
dust analog was used and the same dust flux was simulated as for the DUNE 2009 experiments
described in Desboeufs et al. (2014). To mimic a realistic wet flux event of 10 g m$^{-2}$, 3.6 g of this
analog dust were quickly diluted into 2 L of ultrahigh-purity water (UHP water; 18.2 MΩ cm$^{-1}$
resistivity), and sprayed at the surface of the tanks using an all-plastic garden sprayer (duration =
30 min). Depending on the considered parameter or process, seawater sampling was conducted 1
h (t1h), 6 h (t6h), 12 h (t12h), 24 h (t24h), 48 h (t48h) and 72 h (t72h) (+ 96 h = t96h for station
FAST) after dust addition. Acid-washed silicone tubes were used for transferring the water
collected from the tanks to the different vials or containers.
## 2.2. Stocks
## 2.2.1. Dissolved and particulate organic carbon

The concentration of dissolved organic carbon (DOC) was determined from duplicate 10

mL GF/F (pre-combusted , Whatman) filtered subsamples that were transferred to pre-combusted
glass ampoules, acidified with $H_3PO_4$ (final pH = 2) and sealed. The sealed glass ampoules were
stored in the dark at room temperature until analysis at the Laboratoire d'Océanographie
Microbienne (LOMIC). DOC measurements were performed on a Shimadzu© TOC-V-CSH





(Benner and Strom, 1993). Prior to injection, DOC samples were sparged with $CO_2$-free air for 6
min to remove inorganic carbon. Sample (100 µL) were injected in triplicate and the analytical
precision was 2%. Standards were prepared with acetanilid.
Seawater samples for measurements of particulate organic carbon concentrations (POC; 2
L) were taken at t-12h, t0, t12h, t24h, t48h and t72h (or t96h for station FAST), filtered on pre-
combusted GF/F membranes, dried at 60 °C and analyzed at the Laboratoire d'Océanographie de
Villefranche (LOV, France) following decarbonatation with a drop of HCl 2N, on an elemental
analyzer coupled with an isotope ratio mass spectrometer (EA-IRMS; Vario Pyrocube-Isoprime
100, Elementar©).

## 2.2.2. Total hydrolysable carbohydrates and amino acids

For total hydrolysable carbohydrates and amino acids, samples were taken at t0, t6h,
t24h, t48h and t72h at all stations. For total hydrolysable carbohydrates (TCHO) > 1 kDa,
samples (20 mL) were filled into pre-combusted glass vials (8 h, 500 °C) and stored at -20 °C
pending analysis. Prior to analysis, samples were desalted with membrane dialysis (1 kDa
MWCO, Spectra Por) at 1 °C for 5 h. Samples were subsequently hydrolyzed for 20 h at 100 °C
with 0.8 M HCl final concentration followed by neutralization using acid evaporation ($N_2$, for 5
h at 50 °C). TCHO were analysed at GEOMAR using high performance anion exchange
chromatography with pulsed amperometric detection (HPAEC-PAD), on a Dionex ICS 3000 ion
chromatography system following the procedure of Engel and Händel (2011). Two replicates per
TCHO sample were analyzed.
For total hydrolysable amino acids (TAA), samples (5 mL) were filled into pre-
combusted glass vials (8 h, 500 °C) and stored at -20 °C. Samples were hydrolyzed at 100 °C for





20 h with 1 mL 30% HCl (Suprapur®, Merck) added to 1 mL of sample, and neutralized by acid
evaporation under vacuum at 60 °C in a microwave. Samples were analyzed by high
performance liquid chromatography (HPLC) using an Agilent 1260 HPLC system following a
modified version of established methods (Dittmar et al., 2009; Lindroth and Mopper, 1979).
Separation of 13 amino acids with a C18 column (Phenomenex Kinetex, 2.6 μm, 150 x 4.6 mm)
was obtained after in-line derivatization with o-phthaldialdehyde and mercaptoethanol. A
gradient with solvent A containing 5 % acetonitrile (LiChrosolv, Merck, HPLC gradient grade)
in sodium dihydrogenphosphate (Suprapur®, Merck) buffer (pH 7.0) and solvent B being
acetonitrile was used for analysis. A gradient from 100% solvent A to 78% solvent A was
produced in 50 min. Two replicates per TAA sample were analyzed.

## 214    2.2.3. Transparent exopolymer particles

Samples for transparent exopolymer particles (TEP) were taken at t0, t24h and t72h at all
stations. The abundance and area of TEP were microscopically measured following the
procedure given in Engel (2009). Samples of 10-50 mL were directly filtered under low vacuum
(< 200 mbar) onto a 0.4 μm Nucleopore membrane (Whatman©) filter, stained with 1 mL Alcian
Blue solution (0.2 g l$^{-1}$ w/v) for 3 s and rinsed with MilliQ water. Filters were mounted on
Cytoclear© slides and stored at -20 °C until analysis. Two filters per sample with 30 images each
were analyzed using a Zeiss Axio Scope.A1 (Zeiss©) and an AxioCam MRc (Zeiss©). The
pictures with a resolution of 1388 x 1040 pixels were saved using AxioVision LE64 Rel. 4.8
(Zeiss©). All particles larger than 0.2 μm$^2$ were analyzed. ImageJ© and R were subsequently
used for image analysis (Schneider, Rasband and Eliceiri 2012, R Core Team, 2014). Filters
prepared with 10 mL MilliQ water instead of samples served as a blank. The carbon content of
TEP (TEP-C) was estimated after Mari (1999) using the size-dependent relationship:



$TEP\text{-}C = a\ \Sigma_i\ n_i\ r_i^D$                                                          (1)
with $n_i$ being the number of TEP in the size class i and $r_i$ being the mean equivalent spherical
radius of the size class. The constant $a = 0.25 * 10^{-6}$ (µg C) and the fractal dimension of
aggregates $D = 2.55$ were used as proposed by Mari (1999). To relate to organic carbon
concentration in seawater, data for TEP-C are given as µmol L$^{-1}$.

## 2.3. Processes


## 2.3.1. Dissolved and particulate $^{14}$C incorporation rates


The photosynthetic production of particulate (< 0.2-2 µm and > 2 µm size fractions) and

dissolved organic matter was determined from samples taken at t0, t24h, t48h and t72h (or t96h
at station FAST) with the $^{14}$C-uptake technique. From each tank, four polystyrene bottles (70
mL; three light and one dark bottles) were filled with sampled seawater and amended with 40
µCi of NaH$^{14}$CO$_3$. Bottles were incubated for 8 h in two extra 300 L tanks maintained under
similar light and temperature regimes than in the experimental tanks (ambient temperature for
C1, C2, D1 and D2 and ambient temperature + 3 °C for G1 and G2). Incubations were
terminated by sequential filtration of the sample through polycarbonate filters (pore sizes 2 µm
and 0.2 µm, 47 mm diameter) using low-pressure vacuum. Filters were exposed for 12 h to
concentrated HCl fumes to remove non-fixed, inorganic $^{14}$C, and then transferred to 4 mL plastic
scintillation vials to which 3.5 mL of scintillation cocktail (Ultima Gold XR, Perkin Elmer©)
were added. For the measurement of dissolved primary production, a 5 mL aliquot of each
sampling bottle was filtered, at the end of incubation, through a 0.2 µm polycarbonate filter (25
mm diameter). This filtration was conducted, under low-pressure vacuum, in a circular filtration



manifold that allows the recovery of the filtrate into 20 mL scintillation vials. The filtrates were
acidified with 200 μL of 50% HCl and maintained in an orbital shaker for 12 h. Finally, 15 mL
of liquid scintillation cocktail was added to each sample. All filter and filtrate samples were
measured onboard in a liquid scintillation counter (Packard© 1600 TR). [14]C-based production
rates (PP; in μg C L$^{-1}$ h$^{-1}$) were calculated as:
$$PP = C_T \times \left( \frac{DPM_{sample} - DPM_{dark}}{DPM_{added} \times t} \right)$$    (2)
where $C_T$ is the concentration of total dissolved inorganic carbon (μg C L$^{-1}$), $DPM_{sample}$ and
$DPM_{dark}$ are the radioactivity counts in the light and dark bottle, respectively, $DPM_{added}$ is the
radioactivity added to each sample, and t is the incubation time (h).
The percentage extracellular release (PER%) was calculated as:
$$PER\% = \frac{PPd}{PPd + PPp} \times 100$$    (3)
where PPd refers to [14]C-based dissolved production and PPp refers to [14]C-based particulate
production (sum of < 2 and > 2 μm size fractions).

## 2.3.2. Integrated [13]C incorporation

Addition of [13]C-bicarbonate (NaH[13]CO$_3$ 99%; Sigma-Aldrich©) was performed in each

tank before t0 in order to increase the isotopic level ($\delta^{13}$C signature) of the dissolved inorganic
carbon pool to ca. 350‰. We followed with time the evolution of the $\delta^{13}$C signature in dissolved
inorganic carbon ($\delta^{13}$C-$C_T$), dissolved organic carbon ($\delta^{13}$C-DOC) and particulate organic carbon
pools ($\delta^{13}$C-POC). For the analysis of the actual $\delta^{13}$C-$C_T$, 60 mL of sampled seawater (at t-12h,
t0, t12h, t24h, t48h and t72h; + t96h at station FAST) was gently transferred to glass vials



avoiding bubbles. Vials were sealed after being poisoned with 12 μL saturated $HgCl_2$ and stored
upside-down at room temperature in the dark pending analysis. At the University of Leuven, a
helium headspace (5 mL) was created in the vials and samples were acidified with 2 mL of
phosphoric acid ($H_3PO_4$, 99%). Samples were left to equilibrate overnight to transfer all $C_T$ to
gaseous $CO_2$. Samples were injected in the carrier gas stream of an EA-IRMS (Thermo©
EA1110 and Delta V Advantage), and data were calibrated with NBS-19 and LSVEC standards
(Gillikin and Bouillon, 2007).

At the same frequency than for $\delta^{13}C$-$C_T$, samples for $\delta^{13}C$-DOC were filtered online (see

above), transferred to 40 mL pre-cleaned borosilicate amber EPA vials with septa caps (PTFE-
lined silicone) and stored in the dark pending analysis at the Ján Veizer Stable Isotope
Laboratory (Ottawa, Canada).

At t-12h, t0, t12h, t24h, t48h and t72h (or t96h at station FAST), the $\delta^{13}C$-POC was

obtained based on the same measurements as described above for POC, on a an elemental
analyzer coupled with an isotope ratio mass spectrometer (EA-IRMS; Vario Pyrocube-Isoprime
100, Elementar©).

Carbon isotope data are expressed in the delta notation (δ) relative to Vienna Pee Dee

Belemnite (VPDB) standard. The carbon isotope ratio was calculated as:
$$R_{sample} = \left( \frac{\delta^{13}C_{sample}}{1000} + 1 \right) \times R_{VPDB} \tag{4}$$
with $R_{VPDB} = 0.011237$.



## 2.3.2. Community metabolism (oxygen light-dark method)


At the same frequency as for $^{14}$C incorporation, from each tank, a volume of 2 L was
sampled in plastic bottles and distributed in 15 biological oxygen demand (BOD; 60 mL)
borosilicate bottles. Five BOD bottles were immediately fixed with Winkler reagents (initial $O_2$
concentrations), five BOD bottles were incubated in the dark for the measurement of community
respiration (CR) in two incubators maintained respectively at ambient temperature for C1, C2,
D1 and D2 and at ambient temperature + 3 °C for G1 and G2. Additionally, five BOD bottles
were incubated for the measurement of net community production (NCP) in the same tanks as
described above for $^{14}$C-incorporation. Upon completion of the incubations (24 h), samples were
fixed with Winkler reagents. Within one day, $O_2$ concentrations were measured using an
automated Winkler titration technique with potentiometric endpoint detection. Analyses were
performed on board with a Metrohm© Titrando 888 and a redox electrode (Metrohm© Au
electrode). Reagents and standardizations were similar to those described by Knap et al. (1996).
NCP and CR were estimated by regressing $O_2$ values against time, and CR was expressed as
negative values. Gross primary production (GPP) was calculated as the difference between NCP
and CR. The combined standard errors were calculated as:
$$SE_{xy} = \sqrt{SE_x^2 + SE_y^2} \tag{5}$$



### 2.3.4. Heterotrophic prokaryotic production and ectoenzymatic activities

At all sampling times, heterotrophic bacterial production (BP, *sensus stricto* referring to heterotrophic prokaryotic production) was determined onboard using the microcentrifuge method with the $^3$H- leucine ($^3$H-Leu) incorporation technique to measure protein production (Smith and Azam, 1992). The detailed protocol is in Van Wambeke et al. (2020b). Briefly, triplicate 1.5 mL samples and one blank were incubated in the dark for 1-2 h in two thermostated incubators maintained respectively at ambient temperature for C1, C2, D1 and D2 and at ambient temperature +3 °C for G1 and G2. Incubations were ended by the addition of TCA to a final concentration of 5%, followed by three runs of centrifugation at 16000 g for 10 min. Pellets were rinsed with TCA 5% and ethanol 80%. A factor of 1.5 kg C mol leucine$^{-1}$ was used to convert the incorporation of leucine to carbon equivalents, assuming no isotopic dilution (Kirchman et al., 1993).

Ectoenzymatic activities were measured fluorometrically, using fluorogenic model substrates that were L-leucine-7-amido-4-methyl-coumarin (Leu-MCA) and 4 methylumbelliferyl – phosphate (MUF-P) to track aminopeptidase activity (LAP) and alkaline phosphatase activity (AP), respectively (Hoppe, 1983). Stocks solutions (5mM) were prepared in methycellosolve and stored at -20 °C. Release of the products of LAP and AP activities, MCA and MUF, were followed by measuring increase of fluorescence (exc/em 380/440 nm for MCA and 365/450 nm for MUF, wavelength width 5 nm) in a VARIOSCAN LUXmicroplate reader calibrated with standards of MCA and MUF solutions. For measurements, 2 mL of unfiltered samples from the tanks were supplemented with 100





µL of a fluorogenic substrate solution diluted so that different concentrations were
dispatched in a black 24-well polystyrene plate in duplicate (0.025, 0.05, 0.1, 0.25, 0.5, 1 µM
for MUF-P, 0.5, 1, 5, 10, 25 µM for MCA-leu). Incubations were carried out in the same
thermostatically controlled incubators than those used for BP and reproducing temperature
levels in the experimental tanks. Incubations lasted up to 12 h long with a reading of
fluorescence every 1 to 2 h, depending on the intended activities. The rate was calculated
from the linear part of the fluorescence versus time relationship. Boiled-water blanks were
run to check for abiotic activity. From varying velocities obtained, we determined the
parameters Vm (maximum hydrolysis velocity) and Km (Michaelis-Menten constant which
reflects enzyme affinity for the substrate) by fitting the data using a non-linear regression on
the following equation:
$$V = V_m \text{ x} \frac{S}{K_m + S} \qquad\qquad (6)$$
where V is the hydrolysis rate and S the fluorogenic substrate concentration added.

## 339  2.3.5. Inorganic and organic material export

At the end of each experiment (t72h for TYR and ION and t96 h for FAST, after artificial
dust seeding), the sediment traps were removed, closed and stored with formaldehyde 4%. Back
in the laboratory, after the swimmers were removed, the samples were rinsed to remove the salts
and then freeze-dried. The total amount of material collected was first weighted to measure the
total exported flux. Several aliquots were then weighted to measure the following components:
total carbon and organic carbon, lithogenic and biogenic silicates and calcium. Total carbon was
measured on an elemental analyzer coupled with an isotope ratio mass spectrometer (EA-IRMS;





347 Vario Pyrocube-Isoprime 100, Elementar©). Particulate organic carbon (POC) was measured in

348 the same way after removing inorganic carbon by acidification with HCl 2N. Particulate

349 inorganic carbon (PIC) was obtained by subtracting particulate organic carbon from particulate

350 total carbon. Calcium concentrations were measured by ICP-OES (Inductively Coupled Plasma -

351 Optic Emission Spectrometry; Perkin-Elmer© Optima 8000) on acid digested samples (the

352 organic matrix was removed by $HNO_3$ while the mineral aluminosilicate matrix was eliminated

353 with HF). Biogenic silica (BSi) and Lithogenic silica (LSi) were measured by colorimetry

354 (Analytikjena© Specor 250 plus spectrophotometer) after a NaOH/HF digestion, respectively

355 (Mosseri et al., 2005). The carbonate fraction of the exported material was determined from

356 particulate calcium concentrations (%$CaCO_3$ = 5/2 x (%Ca). The organic matter fraction was

357 calculated as 2 x (%POC). The lithogenic fraction was calculated as [total mass – (organic matter

358 + $CaCO_3$ + opal) and was very comparable to the lithogenic fraction calculated from LSi (taking

359 Si concentration in dust analog used for seeding from Desboeufs et al., 2014; ca. 11.9%). In the

360 controls, the amount of material exported was low and the entire content of the traps was filtered

361 in order to measure total mass and organic matter mass fluxes.

## 363 2.4. Data processing

364  All metabolic rates were integrated over the duration of the experiments using trapezoidal

365 integrations and the relative changes (in %) in tanks D and G as compared to the controls

366 (average between C1 and C2) were computed following:

367 Relative change $= \left( \frac{\text{Rate}_{\text{Treatment}} - \text{Rate}_{\text{Controls}}}{\text{Rate}_{\text{Controls}}} \right) \times 100$    (7)

368 Where $\text{Rate}_{\text{Treatment}}$ is the integrated rate measured in treatments D and G (D1, D2, G1 or G2) and

369 $\text{Rate}_{\text{Controls}}$ is the averaged integrated rates between the duplicate controls (treatment C). Daily





rates of [14]C-based production were computed from hourly rates assuming a 14 h daylight period.
As incubations performed from samples taken at t0 (before dust addition) do not represent what
happened in the tanks between t0 and t24h, as a first assumption, we considered a linear
evolution between these rates and those measured from samples at t24h, and recomputed an
average value for the time interval t0 - t24 h. At FAST, no incubations were performed for [14]C
incorporation and oxygen metabolism between t72h and t96h, again an average rate between
rates measured from samples taken at t48h and t96h was used for this time interval. Since
bacterial respiration rates were not measured, bacterial growth efficiency (BGE, expressed as a
percentage) was estimated based on BP (carbon units) and community respiration (CR, oxygen
units). As BP was determined more often than CR during the first 48 h, hourly BP rates were
integrated using trapezoidal integrations during the time period when CR was measured. We
assumed that heterotrophic prokaryotes were responsible for 70% of CR (BR/CR ratio; Lemée et
al., 2002) and used a respiratory quotient (RQ) of 0.8 (del Giorgio and Williams, 2005),
following the equation:
$$BGE = \left( \frac{BP}{CR \times \frac{BR}{CR}ratio \times RQ + BP} \right) \times 100 \tag{8}$$
When BP varied following an exponential growth, we calculated growth rates ($\mu_{BP}$) from linear
least square regression of ln BP rates versus time.



## 3. Results

### 3.1. Initial conditions

Initial conditions in terms of the chemical and biological standing stocks measured while filling the tanks at the three stations are fully described in Gazeau et al. (2020). Briefly, the three experiments were conducted with surface seawater collected during stratified oligotrophic conditions typical of the open Mediterranean Sea at this period of the year (Table 1). Nitrate + nitrite ($NO_x$) concentrations were maximal at station FAST with a $NO_x$ to dissolved inorganic phosphate (DIP) molar ratio of ~ 4.6. Very low $NO_x$ concentrations were observed at stations TYR and ION (14 and 18 nmol $L^{-1}$, respectively). DIP concentrations were the highest at station TYR (17 nmol $L^{-1}$) and the lowest at the most eastern station (ION, 7 nmol $L^{-1}$). Consequently, the lowest $NO_x$:DIP ratio was measured at TYR (0.8), compared to ION and FAST (2.8 and 4.6, respectively). Silicate ($Si(OH_4)$) concentrations were similar at TYR and ION (~1 μmol $L^{-1}$) and the lowest at FAST (~0.6 μmol $L^{-1}$). Both POC and DOC concentrations were the highest at station TYR (12.9 and 72.2 μmol $L^{-1}$, respectively) and the lowest at FAST (6.0 and 69.6 μmol $L^{-1}$, respectively). Very low and similar concentrations of chlorophyll *a* were measured at the three stations (0.063 - 0.072 μg $L^{-1}$). Phytoplankton communities at stations TYR and ION were dominated by Prymnesiophytes followed by Cyanobacteria, while, at station FAST, the phytoplanktonic community was clearly dominated by photosynthetic prokaryotes. At all three stations, the proportion of pigments representative of larger species was very small (< 5%; Gazeau et al., 2020). Heterotrophic prokaryotes were the most abundant at station FAST (6.15 x $10^5$ cells $mL^{-1}$) and the least abundant at station ION (2.14 x $10^5$ cells $mL^{-1}$).



Relatively similar $^{14}$C-based particulate production rates were measured at the start of the
experiments (t0) in the control tanks (C1 and C2) at station ION and FAST (ca. 0.014 - 0.015 µg
C L$^{-1}$ h$^{-1}$). At both stations, ca. 80% of the production was attributed to larger (> 2 µm) cells and
the percentage of extracellular release (%PER) did not exceed 45%. Lower rates were estimated
at station TYR (total particulate production of 0.08 µg C L$^{-1}$ h$^{-1}$) from which 87.5% was due to
large cells > 2 µm. A larger amount of $^{14}$C incorporation was released as dissolved organic
matter at station TYR compared to the two other stations (PER ca. 60%). Metabolic balance
derived from oxygen measurements showed that, at all three stations, the community was net
heterotrophic with a higher degree of heterotrophy at station TYR (NCP were -1.9, -0.2, -0.8
µmol O$_2$ L$^{-1}$ d$^{-1}$ at TYR, ION and FAST, respectively, as measured in the controls from seawater
sampled at t0). CR and GPP rates were respectively the highest and the lowest at station TYR
compared to the other two stations. Finally, BP rates were the highest at station FAST (35.8 ng C
L$^{-1}$ h$^{-1}$), intermediate at ION (26.1 ng C L$^{-1}$ h$^{-1}$) and the lowest at TYR (21.3 ng C L$^{-1}$ h$^{-1}$).

## 421  3.2. Changes in biological stocks

DOC concentrations showed a general increasing trend during the three experiments and
a large variability between duplicates (Fig. 2). This variability appeared as soon as 1 h after dust
seeding (t1h) while the range of variation at t0 (before dust seeding) was rather moderate
(difference between minimal and maximal values in all tanks of 1.3, 6.2 and 4.3 µmol C L$^{-1}$ at
station TYR, ION and FAST, respectively). As a consequence of this variability, no clear impact
of dust seeding (D) could be highlighted at station TYR and FAST. Indeed, DOC concentrations
in the two duplicates (D1 and D2) were higher than values in the controls (C1 and C2) in only
33% of the samples along the experiments (after dust seeding). In contrast, at station ION, DOC
concentrations appeared impacted by dust seeding as higher concentrations were almost





systematically (83% of the time after dust seeding) measured for this treatment as compared to
control tanks at the same time. At all stations, this impact was somewhat exacerbated under
conditions of temperature and pH projected for 2100 (G1 and G2) as DOC concentrations were
almost all the time higher in these tanks than in control tanks (83 - 100% of the samples after
dust seeding, depending on the station).
Total hydrolysable carbohydrates and amino acids concentrations along the three
experiments are shown in Fig. S2. TCHO concentrations were quite variable between tanks
before dust seeding (t0; 649 - 954, 569 - 660 and 600 - 744 nmol L$^{-1}$ at station TYR, ION and
FAST, respectively) and no visible impact of the treatments were visible at station TYR (TCHO
tended to decrease everywhere). In contrast, at station ION and FAST, values in dust amended
tanks increased and appeared higher than in control tanks towards the end of the experiments
although the large variability between duplicates tended to mask this potential effect. An impact
of dust seeding was much clearer for TAA concentrations that showed larger increases
throughout the three experiments in tanks D1 and D2 as compared to control tanks, this effect
being exacerbated for warmer and acidified tanks (G1 and G2). The ratio between TAA and
DOC concentrations (Fig. 2) showed increasing trends in tanks D and G during all three
experiments with a clear distinction between treatments at the end of the experiments (G > D >
C). The strongest increase was observed at station FAST in tanks G where final values were
above 3%.
Particulate organic carbon (POC) concentrations strongly decreased at all stations
between t-12h and t0, this decrease being the largest at station TYR where concentrations
dropped from 25.7 to 9.6 - 13.2 µmol C L$^{-1}$ (Fig. 3). After dust seeding, POC concentrations did
not show clear temporal trends for the three experiments although a slight general increase could



be observed at station FAST. Furthermore, no impact of dust seeding and warming/acidification
could be observed for this parameter. While concentrations of transparent exopolymer particles
(TEP-C) were rather constant through time in control tanks at the three stations, a large increase
was observed in dust-amended tanks (D and G) with TEP-C reaching values up to ~2 μmol C L$^{-1}$
in tank G1 at station TYR after 24 h (i.e. ~17% of POC concentration, Fig. 3). In all cases except
for tank G2 at station ION, TEP-C further decreased towards the end of the experiments although
concentrations remained well above those observed in the controls. As the variability between
duplicated tanks G was rather high, no impact of warming/acidification on TEP dynamics could
be highlighted at the three stations.

## 3.3. Changes in metabolic rates


$^{14}$C-based particulate production rates as measured during the different time intervals at
the three stations were low in control tanks (maximal total particulate production of 0.34 μg L$^{-1}$
h$^{-1}$ at station FAST) and did not show any particular temporal dynamics (Fig. 4). In these tanks,
the vast majority of particulate production was attributed to cells above 2 μm (65 - 89%). The
percentage of extracellular release (%PER) was overall maximal at station TYR and minimal at
station FAST with a tendency to decrease with time at the three stations although large variations
were observed between duplicates.
Dust addition alone did not have any clear positive impact on all $^{14}$C-based rates at
station TYR, with even an observable decrease in production rates from larger cells (> 2 μm)
compared to the controls. In contrast, at this station, dust seeding under warmer and acidified
conditions (tanks G) had a positive effect on particulate production rates, this effect being
particularly visible for cells < 2 μm and to a lesser extent on dissolved production with a general



decrease of %PER. An important discrepancy between the duplicates of treatment G was
observable at the end of the experiment with much larger rates measured in tank G2.

In contrast to station TYR, an enhancement effect of dust addition was clearly visible at

station ION where all rates increased towards the end of this experiment reaching a maximal
total particulate production of 0.6 - 0.7 µg L$^{-1}$ h$^{-1}$ in tanks D1 and D2. Since this positive effect
was similar between small and larger cells, dust addition alone had no effect on the partitioning
of production at this station, with cells > 2 µm representing ~80% of total production. Although
being also positively impacted and increasing with time, dissolved production appeared less
sensitive than particulate production leading to an overall decrease of %PER at this station
following dust addition. These positive impacts of dust seeding on $^{14}$C-based particulate
production rates were even more visible at this station under warmer and acidified conditions
(tanks G) with maximal rates more than doubled compared to those measured under present
conditions of temperature and pH (1.5 - 1.6 µg L$^{-1}$ h$^{-1}$). Dust seeding under warmer and acidified
conditions had a slight impact on the partitioning of particulate production at this station with
smaller cells benefiting the most from these conditions. %PER remained between 20 and 30%.

At station FAST, similarly to station ION, total particulate production rates were clearly

enhanced by dust addition (tanks D) reaching maximal values during the incubation time interval
t48 - 56h. No clear increase was observed for total particulate production on the next incubation
(t96 - 120h) while production rates of cells larger than 2 µm increased and rates of smaller cells
decreased. However at FAST, in contrast to station ION, there was much less impact of
warming/acidification on all measured rates although rates measured on smaller cells (< 2 µm)
did not decrease at the end of the experiment as observed under present environmental



conditions. %PER under both present conditions of temperature and pH (tanks D) decreased
during this experiment reaching values lower than in the controls and in tanks G.

The initial enrichment of the tanks in $^{13}$C-bicarbonate led to an increase in the $^{13}$C

signature of dissolved inorganic carbon ($\delta^{13}$C-$C_T$) of above 300‰, with generally lower values
measured in warmer and acidified tanks (G; Fig. S3). After this initial enrichment, $\delta^{13}$C-$C_T$ levels
decreased linearly in all tanks. At stations TYR and ION, the isotopic signature of dissolved
organic carbon ($\delta^{13}$C-DOC; Fig. S3) increased with time, although these increases were rather
low and limited to ~ 4‰ over the course of the experiments. In contrast to station TYR, at ION,
an enhanced incorporation of $^{13}$C into DOC was visible after 24 h in tanks D and G in
comparison to control tanks. A similar observation was done at station FAST, especially at the
end of the experiment, although much more variability was observed at this station.

The incorporation of $^{13}$C onto particulate organic carbon ($\delta^{13}$C-POC) is shown in Fig. 5.

At all stations, $\delta^{13}$C-POC increased with time but reached lower enrichment levels at station
TYR as compared to ION and FAST. At this station, incorporation rates appeared smaller in
dust-amended tanks under present environmental conditions (tanks D). As for $^{14}$C-based
production rates, an important discrepancy was observed between duplicates under future
conditions of temperature and pH (tanks G) with much higher final $\delta^{13}$C-POC at the end of the
experiment in tank G2. At station ION, enrichment levels obtained at the end of the experiment
were more important in dust-amended tanks reaching maximal levels of 73‰ in tank G2 at t72h.
This enhancement effect was even more visible at station FAST with maximal enrichment levels
of 146‰ (tank D2 at t96h). Since no sampling occurred at t72h, these enrichment levels cannot
be directly compared to what was measured at station TYR and ION. However, by interpolating





values at t72h assuming a linear increase between these time intervals, enrichment levels
appeared similar although slightly higher for tanks D between station ION and FAST.

NCP rates as measured using the $O_2$ light-dark method showed that, under control

conditions, the communities remained the vast majority of the time throughout the three
experiments in a net heterotrophic state (NCP < 0; Fig. 6). This was especially true at station
TYR where the lowest NCP rates were measured. At this station, dust addition whether under
present or future conditions of temperature and pH did not switch the community towards net
autotrophy but even drove the community towards a stronger heterotrophy. This was related to
the fact that while gross primary production rates were not positively impacted, community
respiration increased in tanks D and G. At station ION, dust addition alone (tanks D) led to a
switch from net heterotrophy to net autotrophy after two days of incubation due to a stronger
positive effect of dust on GPP than on CR. Under future environmental conditions (tanks G), the
same observation was made with higher NCP and GPP rates than in tanks D. CR rates reacted
quickly to these forcing factors in tanks G and initially (first incubation) drove the community
towards a much stronger heterotrophy as compared to the other tanks. Finally, at station FAST,
similarly to what was observed at ION, the community became autotrophic after two days of
incubation in dust amended tanks as, although both GPP and CR were positively impacted by
dust addition, this impact was less important for CR. Warming and acidification had a limiting
impact on this enhancement, with a lower final NCP in tanks G compared to tanks D, a
difference that can be related to an absence of effects of these environmental stressors on GPP
while CR clearly increased at higher temperature and lower pH.

While BP remained constant or gradually increased in control tanks depending on the

station, a clear and quick fertilization effect was observable following dust addition (treatment D



and G) at all stations (Fig. 7). At station TYR, BP rates sharply increased to reach maximal
values at t24h, with an even stronger increase observed under warmer and acidified conditions
(tanks G). After this initial increase, rates slightly decreased towards the end of the experiment.
This fertilization effect appeared less important at station ION where lower maximal rates were
obtained after 24 h as compared to station TYR. Nevertheless, the same observations can be
made, namely, 1) higher rates were measured under future temperature and pH levels and 2) after
this initial sharp increase, rates gradually decreased towards the end of the experiment especially
in tanks G. At station FAST, a much stronger effect of warming/acidification was observed with
an important increase of BP in tanks G until 24 or 48 h post-seeding, depending on the duplicate.
A sharp decline was observed for this treatment until the end of the experiment although rates
remained higher than those measured in tanks C and D. The impact of dust addition under
present environmental conditions (tanks D) was somehow more limited than at the other stations
with a gradual increase until t72h with maximal rates ~ 40 - 100% higher than rates measured in
the controls. However, BP increased exponentially between t0 and t12h in all tanks including
controls, and in all experiments (Table 2). The growth rate of BP ($\mu_{BP}$) in control tanks was the
highest at TYR, intermediate at ION and the lowest at FAST. $\mu_{BP}$ increased significantly in all
dust amended tanks compared to controls. Under future environmental scenarios, $\mu_{BP}$ tended to
increase compared to treatment D but with a variable relative change.

BGE increased in dust amended tanks under present environmental conditions (treatment

D) at TYR and ION, while no changes were detectable at station FAST due to a strong
discrepancy between control duplicates and overall higher BGE at this station in the controls
(Table 3). In contrast, warming and acidification exerted the strongest effect at station FAST
with a doubling of BGE between treatment G and D. Although an increase in BGE was also





observed at the two other stations in treatment G as compared to present environmental
conditions (treatment D), this increase was more limited (ca. 1 to 1.4-fold increase).

The alkaline phosphatase Vm (AP Vm) increased in all experiments after dust seeding,

with amplified effects in G treatments (Fig. S4). Note that AP Vm increased also in the controls
at TYR and FAST. In contrast, leucine aminopeptidase Vm (LAP vm) showed succession of
peaks instead of continuously increasing (Fig. S4). It was higher in dust alone treatment (D) as
compared to the controls at TYR and FAST. A larger variability between duplicates at ION
prevents such an observation. At all stations, maximum velocities were measured under future
environmental conditions (G). Vm being possibly influenced by enzyme synthesis but also by the
number of cells inducing such enzymes, we computed also specific AP Vm per heterotrophic
bacterial cell (Fig. 7). Specific AP Vm slightly increased during all experiments in controls and
dust-amended tanks (D) with no visible differences between these treatments, a clear over-
expression of this enzyme was observed under warmer and more acidified conditions (treatment
G) especially at station FAST where velocities were enhanced by a ~8-fold at t96h.

## 3.4. Inorganic and organic material export

Both total mass and organic matter fluxes, as measured from analyses of the sediment

traps at the end of each experiment, were extremely low under control conditions (Fig. 8).
Additions of dust in tanks D and G led to a strong increase in both fluxes with a large variability
between the duplicates of treatment D at ION. No clear changes between tanks maintained under
present and future conditions of temperature and pH could be highlighted.



## 4. Discussion

## 4.1. Initial conditions of the tested waters and evolution in controls

As discussed in the companion paper from Gazeau et al. (2020), the three sampling stations were typical of stratified oligotrophic conditions encountered in the open Mediterranean Sea in late spring / early summer. DOC concentrations at the start of the experiments were in the same range (60 - 75 µmol C L$^{-1}$) as those measured from samples collected in surface waters using clean sampling procedures (Van Wambeke et al., 2020b), revealing no contamination issues from our sampling device. TAA concentrations as measured in the tanks at t0 were also consistent with measurements from surface water samples (Van Wambeke et al., 2020b) with an average across stations and treatments of $254 \pm 36$ nmol L$^{-1}$ (Fig. S2). In contrast, TCHO appeared higher at t0 (average across stations and treatments of $681 \pm 98$ nmol L$^{-1}$) than concentrations based on clean *in situ* sampling (average of $595 \pm 43$ nmol L$^{-1}$; Van Wambeke et al., 2020b). The decrease in POC concentrations between pumping (t-12h) and t0 for the three experiments, especially at station TYR (likely linked to higher initial concentrations), was likely a consequence of sedimentation of senescent cells and/or fecal pellets in our experimental systems, which are designed to evaluate the export of matter thanks to their conical shape. TEP concentrations were not quantified at t-12h and therefore there is no possibility to evaluate if sedimentation of these particles occurred before t0 in our tanks. At t0, larger and more abundant TEP were measured at station TYR compared to the two other stations (data not shown) leading to a larger contribution of TEP carbon content (TEP-C) to POC concentrations (Fig. 3).





As a consequence of a very low availability in inorganic nutrients, TChl$a$ and $^{14}$C-based
production rates were very low, all typical of oligotrophic conditions. Nano- and micro-
phytoplanktonic cells (> 2 μm) contributed most of the $^{14}$C-based particulate production (~ 80%),
as found also on several on-deck incubations at the three stations (on average $73 \pm 6\%$; Marañón
et al., 2020). %PER values were also very similar to those measured during these on-deck
incubations (~ 40-45%; see Marañón et al., 2020). This suggests no significant impact of our
experimental protocol on rates and partitioning of $^{14}$C-based production rates (i.e. sampling from
the continuous seawater supply, delay of 12 h before initial measurements, artificial light etc.).
The low values of chlorophyll stocks as well as of $^{14}$C-based production rates are consistent with
previous estimates based on direct measurements, satellite observations and modelling
approaches in the same areas in late spring / early summer (e.g. Bosc et al., 2004; Lazzari et al.,
2016; Moutin and Raimbault, 2002).
The metabolic balance was in favor of net heterotrophy at all stations at the start of the
experiments (NCP < 0). Net heterotrophy in the open Mediterranean sea at this period of the year
has been reported by Regaudie-de-Gioux et al. (2009) and Christaki et al. (2011) in agreement
with our measurements at t0 in control tanks (Table 1). The lowest NCP and the highest CR rates
were measured at station TYR, suggesting that the autotrophic plankton community was not very
active at this station. This was confirmed by the $^{14}$C-based particulate production rates, which
were about half the ones measured at the other two stations. The community at TYR was most
likely relying on regenerated nutrients, as shown by the highest levels of ammonium ($NH_4^+$)
measured at the start of this experiment (Gazeau et al., 2020). As discussed in Guieu et al.
(2020), a dust deposition event took place several days before the arrival of the vessel in this
area, likely on May 10-12. This dust event was confirmed by inventory of particulate aluminium





in the water column at several stations of the Tyrrhenian Sea including TYR, 6 to 9 d after the
event (Matthieu Bressac, pers. comm.). This dust deposition likely stimulated phytoplankton
growth and POC accumulation shortly after the deposition and consequently increased the
abundance of herbivorous grazers (copepods) and attracted carnivorous species (Feliú et al.,
2020), subsequently driving the community towards a net heterotrophic state that characterized
the initial condition of the experiment at this station. The optimal conditions for BP growth at
this station were also confirmed by the highest $\mu_{BP}$ growth rates obtained among the three
experiments (Table 2; 0.06 - 0.07 h$^{-1}$) in controls tanks.

The two other stations, although both also showing a slight net heterotrophic state, were

clearly different from each other in terms of initial biological stocks and metabolic rates. Indeed,
whereas TChl$a$ and abundances of pico- and nano-autotrophic cells (flow cytometry counts;
Gazeau et al., 2020) were higher at FAST compared to ION, the autotrophic community was not
more efficient at fixing carbon at this station, as shown by similar initial $^{14}$C-based production
rates. In contrast, both heterotrophic prokaryotic abundances and BP were much higher at station
FAST as compared to ION, leading to initial higher CR and lower NCP. At ION, the initial NCP
closer to metabolic balance further suggests a tight coupling between heterotrophic prokaryotes
and phytoplankton at this station, as discussed by Dinasquet et al. (2021).

For most of the chemical and biological stocks (e.g. nutrients, pigments etc.) presented in

Gazeau et al. (2020), no major changes took place during the three experiments under control
conditions. Here, we further show that DOC, POC as well as TEP concentrations did not exhibit
strong changes during the experiments. For DOC, large variability between the duplicates (C1
and C2) potentially masked an increase towards the end of the experiments. The same holds true
for autotrophic metabolic rates, as $^{14}$C-based particulate production rates showed no marked





variations during the three experiments, although a slight increase was visible at FAST until
t48h. The communities at the three stations remained heterotrophic under the nutrient-limited
conditions in the controls. However, heterotrophic prokaryotes probably benefited from initial
inputs of available organic matter issued from other stressed eukaryotic organisms and/or POC
decay between t-12h and t0, which could be due to both sedimentation and degradation. This was
reflected in the progressive increase of BP, their variable initial growth rates ($\mu_{BP}$ ranged from
0.02 to 0.06 $h^{-1}$ in control tanks according to the experiment) as well as increasing TAA/DOC
ratios at the three stations. Finally, an initial increase of BP during incubations is generally
described and classically attributed to a bottle effect, which favours large, fast-growing bacteria
and often induces mortality of some phytoplankton cells (Calvo-Díaz et al., 2011; Ferguson et
al., 1984; Zobell and Anderson, 1936)

## 4.2. Impact of dust addition under present environmental conditions

The addition of nitrogen and phosphorus in the experimental tanks through dust seeding
(+ 11 to + 11.6 $\mu mol\ L^{-1}$ and + 22 to + 30.8 $nmol\ L^{-1}$ for $NO_x$ and DIP, respectively, in dust
enriched, i.e. D1 and D2, versus controls; Gazeau et al., 2020) had very contrasting impacts on
the metabolism of the communities, depending on the station. At TYR, surprisingly, the relieving
of nutrient limitation had a negative impact on $^{13}C$ incorporation as well as on both particulate
and dissolved $^{14}C$-based production rates (as seen by the relative changes compared to the
control presented in Fig. 9). These observations are fully corroborated by the observed relative
decrease in GPP in these tanks (D1 and D2) relative to controls and by the negative impact of
dust-addition on TChl$a$ concentrations as discussed by Gazeau et al. (2020). Integrated $^{14}C$-





incorporation rates converted to P (using a C:P molar ratio of 245:1 determined in the particulate
organic matter in surface waters of the Northwestern Mediterranean Sea during stratification;
Tanaka et al., 2011) showed that phytoplankton P requirements in treatment D (~2 nmol P L$^{-1}$)
were much lower than the release of DIP through dust addition at this station (+ 20.4 to + 24.6
nmol P L$^{-1}$; Gazeau et al., 2020). This suggests that the observed strong decrease of DIP at this
station following dust addition was due to an utilization by the heterotrophic compartment.
Indeed, in contrast to the autotrophic compartment, both heterotrophic prokaryotic abundances
(Gazeau et al., 2020) and BP (this study, Fig. 9) showed that heterotrophic prokaryotes reacted
quickly and strongly to the increase in DIP availability. Integrated BP increased by almost 400%
in tanks D1 and D2 as compared to controls (Fig. 9). Such relative increases of BP surpassing by
far the observed relative increases of CR suggest a much more efficient utilization of resources
by heterotrophic prokaryotes in this treatment (i.e. BGE increased by 200% as compared to the
controls; Fig. 9). As such, at this station, the addition of dust drove the community to an even
more heterotrophic state. Such absence of response of the autotrophic community despite the
input of new N and P from simulated wet deposition was never observed in dust enrichment
experiments performed in the Mediterranean Sea (Guieu and Ridame, 2020). To the best of our
knowledge, it is the first time that a negative effect of dust addition is experimentally
demonstrated on the metabolic balance. The apparent utilization of nutrients, especially DIP
(Gazeau et al., 2020), by heterotrophic prokaryotes was extremely fast, starting right after dust
addition and driving DIP concentrations back to control levels at the end of the experiment
(t72h). While heterotrophic prokaryotic abundances increased until the end of the experiment,
BP rates increased exponentially during the fist 24h, and then BP reached a plateau.
Heterotrophic prokaryotes appeared limited by nutritive resources although DIP concentrations





were not yet back to their initial level and no relative increase of the AP Vm per cell compared to
the control was observed in these tanks. Independent nutrient experiments showed a direct
stimulation of BP in the dark after addition of DIP (Van Wambeke et al., 2020b), suggesting a
great competition with phytoplankton for DIP utilization at TYR. After 24 h, abundances of
heterotrophic prokaryotes continued to increase while BP stabilized, suggesting a less extent of
lysis and viral control than in the other experiments (abundances of heterotrophic nanoflagellates
decreased; Dinasquet et al., 2021). This limitation of BP was potentially a consequence of
relatively less available access to labile DOC sources, as [14]C-based production rates decreased
relative to the controls at t24h and t48h although BP increased by 200 - 800%. The very tight
coupling between phytoplankton and bacteria at all stations investigated was further confirmed
by the absence of an important [13]C incorporation into DOC (Fig. S3).

At stations ION and FAST, in contrast to TYR, both the autotrophic and heterotrophic

community benefited from dust addition relative to the controls (Fig. 9). Interestingly, while the
relative increase in integrated autotrophic processes (GPP and all [14]C-based production rates)
was more important at FAST than at ION, the opposite was observed for BP. Estimated BGE
values even suggest an absence of response to dust addition at station FAST compared to the
controls. The different (relative) responses of BP at the two stations could be partly explained by
the dynamics of BP in the control tanks as no clear pattern could be observed at ION while a
continuous increase was observed at FAST. As shown by Gazeau et al. (2020), at FAST,
abundances of heterotrophic prokaryotes were much higher at the start of the experiment, further
increased until t48h and then declined until the end of the experiment.

We can rule out a potential limitation of BP from DIP availability at station FAST as DIP

levels remained much higher in tanks D than in the controls (Gazeau et al., 2020). Furthermore,





the amount of maximum DIP reached before its decline compared to TYR and ION showed a
less important direct DIP uptake, suggesting that communities were not as much P limited at
FAST compared to the other stations at the start of the experiment. Finally, no increase of
specific AP Vm was observed in these tanks as compared to the controls (Fig. 7), suggesting no
particular additional needs for AP synthesis per unit cell following dust addition. A potential
explanation resides in the competition between heterotrophic bacteria and phytoplankton for DIP
utilization. At station ION, P requirements of the autotrophic community were low compared to
the initial input of DIP following dust seeding (~9 nmol P $L^{-1}$ as compared to an input of + 22 to
+ 23.3 nmol P $L^{-1}$; Gazeau et al., 2020). In contrast, at FAST, the autotrophic community
consumed a much larger proportion of the initial DIP input (~25 nmol P $L^{-1}$ as compared to an
input of 30.8 - 31.3 nmol P $L^{-1}$) and phytoplankton appeared as a winner for the utilization of
DIP towards the end of the experiment at this station. It seems that heterotrophic bacteria and
phytoplankton were more in a steady state of equilibrium and less stressed at the start of the
experiment at FAST, i.e. phytoplankton abundances showed no decrease between t-12h and t0
and BP did not increase as much as during the other two experiments, suggesting a strong
predation pressure ($\mu_{BP}$ was the lowest of the three experiments: ca. 0.02 $h^{-1}$ in the controls).

The explanation for the observed differential responses of the autotrophic community at

the two stations (FAST > ION) is not evident and further complicated by the fact that the
sampling strategy differed between the two stations (i.e. no sampling at t72h, replaced by a
sampling at t96h). It is however unlikely that this different sampling strategy was responsible for
the different changes in computed integrated autotrophic rates at the two stations. As a maximal
increase in nano-eukaryote abundance was observed at t72h at FAST (followed by a drastic
reduction at t96h; Gazeau et al., 2020), excluding this sampling point in the calculation of





autotrophic metabolic rates would most likely have led to an underestimation of these rates rather
than an overestimation. Furthermore, a similar partitioning of $^{14}$C-based production rates
throughout the two experiments did not provide clear insights on which size-group benefited the
most at station FAST compared to ION. Two non-exclusive explanations could be proposed: (1)
as mentioned above, a less important immediate consumption of DIP by heterotrophic bacteria
leading to a higher availability of new DIP for phytoplankton growth at FAST (+ 31 vs + 22 to +
23 nmol L$^{-1}$ at FAST and ION, respectively; Gazeau et al., 2020) along with (2) the presence of a
potentially more active community at the start of the experiment at FAST with a much higher
contribution from smaller cells (i.e. pico-eukaryotes, *Synechococcus*; Gazeau et al., 2020) that
are well known to be better competitors for new nutrients and that were less stressed at the start
of the experiments (e.g. Moutin et al., 2002).

During both experiments at ION and FAST, communities switched from net heterotrophy

to net autotrophy between 48 and 72 h following dust addition (Fig. 6), leading to a positive
integrated NCP at both stations (Fig. 9). This is an important observation since, to the best of our
knowledge, the present study constitutes the first investigation of the community metabolism
response to dust addition. However, it is important to discuss the timing of such a switch in
community metabolism. Since heterotrophic prokaryotes reacted faster than autotrophs to the
relief of nutrient limitation (i.e. BP already increased by 150-500% at t24 h, while $^{14}$C-based
production rates increased only after 48-72 h), NCP was first lower (and negative) in the dust-
amended tanks as compared to the controls. Marañón et al. (2010) and Pulido-Villena (2008,
2014) have already reported on a much faster response of the heterotrophic prokaryote
community to dust enrichment in the central Atlantic Ocean and Mediterranean Sea,
respectively. As DIP concentrations at the completion of their 48 h incubations did not differ





from that in the controls, it is unlikely that primary production rates and consequently NCP
would have further increased. In contrast, during our experiments, DIP concentrations in dust-
amended tanks (D) reached initial levels only after 72 h at TYR and ION and remained far above
ambient levels at FAST until the end of the experiment (t96h). During the PEACETIME cruise,
high frequency sampling of CTD casts allowed following the evolution of biogeochemical
properties and fluxes before and after wet dust deposition that took place in the area around
FAST on June 3-5 (Van Wambeke et al., 2020a). As in our experiment, a rapid increase in BP
was responsible for the observed *in situ* decline in DIP concentrations in the mixed layer
following the rain with no detectable changes in primary production (Van Wambeke et al.,
2020a). The intensity of the wet deposition event that was simulated during our experiments was,
by far, more important, but still representative of a realistic scenario (Bonnet and Guieu, 2006;
Loÿe-Pilot and Martin, 1996; Ternon et al., 2010).

The most intriguing result concerning the export of inorganic and organic matter is that

these fluxes were maximal at the end of the experiment at TYR in the dust-amended tanks
despite the fact that $^{14}$C-based production was relatively low and not enhanced by dust addition.
Based on previous studies (Bressac et al., 2014; Louis et al., 2017; Ternon et al., 2010), organic
matter export was most likely mainly due to the formation of organic-mineral aggregates
triggered by the introduced lithogenic particles (referred thereafter to as $POC_{litho}$). Indeed, Louis
et al. (2017) showed that such an aggregation process occurs within 1 h after dust deposition.
These authors further demonstrated the key role of TEP as the conversion of dissolved organic
matter (DOM) to POC was mediated by TEP formation/aggregation activated by the introduction
of dust. As TEP concentrations were only measured on two occasions after seeding with the first
measurement occuring at t24h, ), it prevents studying in detail the dynamics of these particles.





Nevertheless, it is very likely that the sharp decrease of TEP abundances (data not shown)
between t24h and t72h was related to $POC_{litho}$ export. The coefficient linking $POC_{litho}$ to $Litho_{flux}$
(i.e. the mass of sedimented particles) measured here (0.02) is consistent with values reported for
other experiments conducted in the Mediterranean Sea (Louis et al., 2017).
Even though $^{14}C$-based production rates were enhanced in the dust-amended tanks at
stations ION and FAST, the amount of POC exported at the end of these experiments remained
lower than at TYR, with fluxes ~ 10-20 mg C m$^{-2}$ d$^{-1}$. It must be stressed that not all the
lithogenic material introduced in the tanks was recovered after 4 (and 5) days, with the highest
percentage (~ 30%) being found at TYR, indicating that the tested waters at this station had a
better capacity to aggregate dust. This efficiency to export $POC_{litho}$ more rapidly at TYR
compared to ION and FAST was likely due to the age and quantity of dissolved organic matter
present at the time of the seeding (Bressac and Guieu, 2013). At TYR, impacted by a strong dust
event several days before the experiment started (see above), the likely stimulation of the
autotrophs after this *in situ* event should have been followed by the production of a fresh and
abundant DOM, comparable to the "post-bloom situation" in Bressac and Guieu (2013).
## 4.3. Impact of dust addition under future environmental
## conditions
Warming and/or acidification had a clear impact on most evaluated stocks and metabolic
rates. Gazeau et al. (2020) have already discussed temperature/pH mediated changes in nutrient
uptake rates and autotrophic community composition in these experiments. The difference in the
relative response of plankton communities to dust addition under present and future conditions of
temperature and pH was highly dependent on the sampling station (Fig. 9). At all stations, $^{14}C$-



based particulate production rates were enhanced under future conditions as compared to those
measured under present environmental conditions (treatment D) although this pattern was not
observed for $^{13}$C incorporation into POC at stations ION and FAST. At ION, no differences
could be detected and at FAST an even lower $^{13}$C-enrichment was measured at the end of the
experiment. These contrasting patterns between $^{14}$C-uptake rates and $^{13}$C-enrichment of POC are
likely explained by the fact that the latter covered the whole experimental period (including dark
periods) and represents net community carbon production while $^{14}$C-based rates were measured
over 8 h incubations in the light, providing an estimate in between gross and net carbon
production.
Similarly, the heterotrophic compartment was more stimulated, as BP rates increased
strongly at all stations under this treatment compared to treatment D. The relatively smaller
increase in CR rates, compared to BP, leading to higher BGE suggests a better utilization of
resources by heterotrophic prokaryotes under future environmental conditions. Overall, CR was
more impacted than GPP, with the consequence that all integrated NCP rates decreased under
future environmental conditions compared to present conditions (treatment D). At station TYR,
as discussed previously, dust addition under present conditions did not lead to a switch from net
heterotrophy to net autotrophy. This pattern was even more obvious under warmer/acidified
conditions, with a larger decrease in integrated NCP at this station. The decrease of integrated
NCP at station FAST relative to controls, as well as the smaller increase of all $^{14}$C-based
production rates relative to those observed at station ION must be taken with caution. As already
discussed, the fact that for these processes (O$_2$ metabolism and $^{14}$C-incorporation), no samples
were taken at FAST at t72h when maximal cell abundances were recorded for all autotrophic
groups (pico- and nano-eukaryotes, autotrophic bacteria) must have artificially led to an





underestimation of these integrated metabolic rates. The question of the timing appeared even
more preponderant under warmer/acidified conditions, especially at station FAST, where the
very important increase in BP led to a full consumption of DIP before t48h (Gazeau et al., 2020)
and drove the community towards a strong heterotrophy. The metabolic balance further switched
to a slight autotrophy at t72h when heterotrophic bacterial activity appeared limited by nutrient
availability.

Both elevated partial pressure of $CO_2$ ($pCO_2$) and warming are major global change

stressors impacting marine communities. Elevated $pCO_2$ may directly facilitate oceanic primary
production through enhanced photosynthesis (Hein and Sand-Jensen, 1997; Riebesell et al.,
2007) although the effects appear to be species- and even strain-specific (e.g. Langer et al.,
2009). Warming affects organisms by enhancing their metabolic rates (Brown et al., 2004;
Gillooly et al., 2001). Although recent studies suggest large differences in temperature sensitivity
between phytoplankton taxa (Chen and Laws, 2017) and no significant overall difference
between algae and protozoa (Wang et al., 2019), mineralization rates are usually believed to be
more impacted by warming than primary production rates, potentially leading to a decline in net
oceanic carbon fixation (Boscolo-Galazzo et al., 2018; Garcia-Corral et al., 2017; Lopez-Urrutia
and Moran, 2007; Regaudie-de-Gioux and Duarte, 2012) and carbon export efficiency (Cael et
al., 2017; Cael and Follows, 2016). Overall, our experimental set-up did not allow discriminating
warming from acidification effects, precluding an evaluation of their potential individual
impacts. Nevertheless, we could speculate to which extent a 3 ºC warming and a doubling of
$CO_2$ can explain some of the observed differences between D and G (for instance, a 2-fold
increase in $^{14}C$-based production rates at ION). For photosynthesis, meta-analysis studies
indicate minor effects of $pCO_2$ on most investigated species (Kroeker et al., 2013; Mackey et al.,





2015). Recent studies show a strong, although species-dependent, temperature sensitivity of
phytoplankton growth (Chen and Laws, 2017; Wang et al., 2019), suggesting that a 3 ºC
warming could explain most of the increased carbon fixation in G compared to D. With respect
to NCP, our results are in line with the general view and suggest a weakening of the so-called
fertilization effect of atmospheric deposition in the coming decades.

In contrast, we did not observe an additional impact of future environmental conditions

on the export of organic matter after dust addition as, at each station, this export was of the same
order of magnitude for treatments D and G. This result is in agreement with the findings of a
similar experiment in coastal Mediterranean waters that considered only pH change (Louis et al.,
2017) but stands in contrast with the findings of Müren et al. (2005) who showed a clear
decrease in sedimentation following a 5 °C warming in the Baltic Sea. Only a few studies have
addressed the combined effect of both temperature and pH changes on aggregation processes and
export but none considered dust as the particulate phase. These studies, focused mainly on the
formation of TEP, were inconclusive on the impact of these combined factors (Passow and
Carlson, 2012, and references therein). As the potential effect of warming and acidification on
biogenic carbon export was certainly, over the rather restricted duration of the experiments,
insignificant as compared to the large amount of carbon exported through the lithogenic pump,
observations over longer temporal scales are probably required to ascertain the interactive effects
of these stressors in the coming decades.





## 5. Conclusion


Although the three experiments were conducted under rather similar conditions in terms

of nutrient availability and chlorophyll stock of the tested seawater, contrasting responses were
observed following the simulation of a wet dust deposition event. Under present conditions of
temperature and pH, at the site where the community was the most heterotrophic (TYR), no
positive impact of new nutrients could be observed on autotrophs, while a fast and strong
response of heterotrophic bacteria drove the metabolic balance towards an even more
heterotrophic state. The situation was different at the two other stations where a more active
autotrophic community responded quickly to the relief in nutrient limitation, driving the
community to an autotrophic state at the end of these experiments. In all tested waters, an overall
faster response of the heterotrophic prokaryote community, as compared to the autotrophic
community, was observed after new nutrients were released from dust. Phytoplankton could
benefit from nutrient inputs, only if the amount released from dust was enough to sustain both
the fast bacterial demand and the delayed one of phytoplankton. As our experimental protocol
consisted in simulating a strong, although realistic, wet dust deposition, further work should
explore at which flux a wet dust deposition triggers an enhancement of net community
production and therefore increases the capacity of the surface oligotrophic ocean to sequester
atmospheric $CO_2$. This question, of the utmost importance in particular for modelling purposes,
should be answered through future similar experiments as the ones considered in our study but
following a gradient approach of dust fluxes. As a consequence of a stronger sensitivity of
heterotrophic prokaryotes to temperature and/or pH, the ongoing warming and acidification of
the surface ocean will result in a decrease of the dust fertilization of phytoplankton in the coming
decades and a weakening the $CO_2$ sequestration capacity of the surface oligotrophic ocean.



## Data availability


All data and metadata will be made available at the French INSU/CNRS LEFE CYBER database
(scientific coordinator: Hervé Claustre; data manager, webmaster: Catherine Schmechtig).
INSU/CNRS LEFE CYBER (2020)

## Author contributions


FG and CG designed and supervised the study. All authors participated in sample analyses. FG
wrote the paper with contributions from all authors.

## Financial support


This study is a contribution to the PEACETIME project (http://peacetime-project.org), a joint
initiative of the MERMEX and ChArMEx components supported by CNRS-INSU, IFREMER,
CEA, and Météo-France as part of the programme MISTRALS coordinated by INSU.
PEACETIME was endorsed as a process study by GEOTRACES and is a contribution to IMBER
and SOLAS International programs. PEACETIME cruise (https://doi.org/10.17600/17000300).
The project leading to this publication has received funding from European FEDER Fund under
project 1166-39417. The research of EM and MPL was supported by the Spanish Ministry of
Science, Innovation and Universities through project POLARIS (Grant No. PGC2018-094553B-
I00) and by European Union's H2020 research and innovation programme through project
TRIATLAS (Grant No. 817578). JD was funded by a Marie Curie Actions-International
Outgoing Fellowship (PIOF-GA-2013-629378).



**Acknowledgments**
The authors thank the captain and the crew of the RV "Pourquoi Pas ?" for their professionalism
and their work at sea. Céline Ridame and Kahina Djaoudi are thanked for their help during
sampling, Sophie Guasco and Marc Garel for their help in ectoenzymatic measurements onboard.

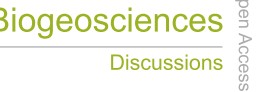

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





# Tables

Table 1. Initial chemical and biological stocks as measured while filling the tanks (initial conditions in pumped surface water; sampling time: t-12h). $NO_x$: nitrate + nitrite, DIP: dissolved inorganic phosphorus, $Si(OH)_4$: silicate, POC: particulate organic carbon, DOC: dissolved organic carbon, TEP: transparent exopolymer particles, TChl$a$: total chlorophyll $a$. Values shown for $^{14}C$ incorporation rates, percentages of extracellular release (%PER) as well as for net community production (NCP), community respiration (CR) and gross primary production (GPP) were estimated from samples taken at t0 in the control tanks. For heterotrophic bacterial production (BP), rates were estimated from seawater sampled at t-12h.

| Sampling station | | TYR | ION | FAST |
|---|---|---|---|---|
| Coordinates (decimal) | | 39.34 N, 12.60 E | 35.49 N, 19.78 E | 37.95 N, 2.90 N |
| Bottom depth (m) | | 3395 | 3054 | 2775 |
| Day and time of pumping (local time) | | 17/05/2017 17:00 | 25/05/2017 17:00 | 02/06/2017 21:00 |
| Temperature (°C) | | 20.6 | 21.2 | 21.5 |
| Salinity | | 37.96 | 39.02 | 37.07 |
| Stocks | $NO_x$ (nmol L$^{-1}$) | 14.0 | 18.0 | 59.0 |
| | DIP (nmol L$^{-1}$) | 17.1 | 6.5 | 12.9 |
| | $Si(OH)_4$ (μmol L$^{-1}$) | 1.0 | 0.96 | 0.64 |






| | | | |
|---|---|---|---|
| POC (μmol L⁻¹) | 12.9 | 8.5 | 6.0 |
| DOC (μmol L⁻¹) | 72.2 | 70.2 | 69.6 |
| TEP (x 10⁶ L⁻¹) | 6.8 | 3.8 | 3.7 |
| TChl$a$ (μg L⁻¹) | 0.063 | 0.066 | 0.072 |
| Heterotrophic prokaryotes abundance (x 10⁵ cell mL⁻¹) | 4.79 | 2.14 | 6.15 |
| Processes ¹⁴C-based total particulate production (μg C L⁻¹ h⁻¹) | 0.08 ± 0.03 | 0.14 ± 0.04 | 0.15 ± 0.04 |
| ¹⁴C-based > 2 μm particulate production (μg C L⁻¹ h⁻¹) | 0.07 ± 0.02 | 0.11 ± 0.02 | 0.11 ± 0.02 |
| ¹⁴C-based < 2 μm particulate production (μg C L⁻¹ h⁻¹) | 0.01 ± 0.01 | 0.04 ± 0.02 | 0.05 ± 0.01 |
| %PER | 60 ± 20 | 45 ± 3 | 32 ± 23 |
| NCP (μmol O₂ L⁻¹ d⁻¹) | -1.9 ± 0.3 | -0.2 ± 0.2 | -0.8 ± 0.9 |
| CR (μmol O₂ L⁻¹ d⁻¹) | -2.6 ± 0.1 | -1.2 ± 0.5 | -1.9 ± 1.6 |
| GPP (μmol O₂ L⁻¹ d⁻¹) | 0.7 ± 0.4 | 1.1 ± 0.3 | 1.1 ± 0.7 |
| BP (ng C L⁻¹ h⁻¹) | 11.6 | 15.2 | 34.6 |






Table 2. Heterotrophic bacterial production (BP) growth rates ($\mu_{BP}$ in h$^{-1}$) estimated from the
exponential phase of BP growth, observable from at least four sampling points, between t0 and
t12h, during the three experiments (TYR, ION and FAST) in the six tanks (controls: C1, C2; dust
addition under present conditions of temperature and pH: D1, D2; dust addition under future
conditions of temperature and pH: G1 and G2). Values ± SE are shown.

|  | $\mu_{BP}$ | | |
|  | TYR | ION | FAST |
| --- | --- | --- | --- |
| C1 | 0.076 ± 0.025 | 0.042 ± 0.007 | 0.020 ± 0.003 |
| C2 | 0.066 ± 0.018 | 0.041 ± 0.005 | 0.026 ± 0.004 |
| D1 | 0.117 ± 0.008 | 0.095 ± 0.020 | 0.089 ± 0.014 |
| D2 | 0.194 ± 0.020 | 0.145 ± 0.007 | 0.090 ± 0.007 |
| G1 | 0.164 ± 0.020 | 0.126 ± 0.011 | 0.124 ± 0.005 |
| G2 | 0.150 ± 0.003 | 0.137 ± 0.033 | 0.163 ± 0.014 |






Table 3. Estimated bacterial growth efficiency (BGE in %) during the course of the three
experiments (TYR, ION and FAST) in the six tanks (controls: C1, C2; dust addition under
present conditions of temperature and pH: D1, D2; dust addition under future conditions of
temperature and pH: G1 and G2). BGE was calculated based on integrated heterotrophic
bacterial production (BP) and community respiration (CR) rates by applying a bacterial
respiration to CR ratio of 0.7 and a respiratory quotient of 0.8 (see Material and Methods).

| | Bacterial growth efficiency (BGE) | | |
|---|---|---|---|
| | TYR | ION | FAST |
| C1 | 11.1 | 9.8 | 15.4 |
| C2 | 11.7 | 14.5 | 22.0 |
| D1 | 31.8 | 21.0 | 17.3 |
| D2 | 32.3 | 30.6 | 19.9 |
| G1 | 39.3 | 35.2 | 37.6 |
| G2 | 32.5 | 34.8 | 38.1 |





**Figure caption**
Fig. 1. Map showing the sampling stations in the Mediterranean Sea along the transect performed
onboard the R/V "Pourquoi Pas ?" during the PEACETIME cruise.
Fig. 2. Dissolved organic carbon (DOC) concentrations and ratio between total hydrolysable
amino acids (TAA) and DOC concentrations measured in the six tanks (controls: C1, C2; dust
addition under present conditions of temperature and pH: D1, D2; dust addition under future
conditions of temperature and pH: G1 and G2) during the three experiments (TYR, ION and
FAST). The dashed vertical line indicates the time of seeding (after t0).
Fig. 3. Particulate organic carbon (POC) concentrations and transparent exopolymer particle
carbon content (TEP-C) measured in the six tanks (controls: C1, C2; dust addition under present
conditions of temperature and pH: D1, D2; dust addition under future conditions of temperature
and pH: G1 and G2) during the three experiments (TYR, ION and FAST). The dashed vertical
line indicates the time of seeding (after t0).
Fig. 4. $^{14}$C-based production rates (< 2 μm and > 2 μm size fractions, total particulate) estimated
from 8 h incubations on samples taken in the six tanks (controls: C1, C2; dust addition under
present conditions of temperature and pH: D1, D2; dust addition under future conditions of
temperature and pH: G1 and G2) during the three experiments (TYR, ION and FAST). The
percentage of extracellular release (%PER) is also shown.
Fig. 5. Incorporation of $^{13}$C into particulate organic carbon ($\delta^{13}$C-POC) in the six tanks (controls:
C1, C2; dust addition under present conditions of temperature and pH: D1, D2; dust addition





under future conditions of temperature and pH: G1 and G2) during the three experiments (TYR,
ION and FAST). The dashed vertical line indicates the time of seeding (after t0).
Fig. 6. Net community production (NCP), community respiration (CR) and gross primary
production (GPP) rates estimated using the oxygen light-dark method (24 h incubations) on
samples taken in the six tanks (C1, C2, D1, D2, G1 and G2) during the three experiments (TYR,
ION and FAST).
Fig. 7. Heterotrophic bacterial production rates (BP) and cell-specific maximum hydrolysis
velocity (Vm) of the alkaline phosphatase (both over 1-2 h incubations) on samples taken in the
six tanks (C1, C2, D1, D2, G1 and G2) during the three experiments (TYR, ION and FAST).
Fig. 8. Total mass and organic matter fluxes measured in the sediment traps at the end of the
three experiments (TYR, ION and FAST) in the six tanks (C1, C2, D1, D2, G1 and G2).
Fig. 9. Relative difference (%) between integrated rates measured in tanks D (D1, D2; dust
addition under present conditions of temperature and pH) and G (G1, G2; dust addition under
future conditions of temperature and pH) as compared to the controls (C1, C2) during the three
experiments (TYR, ION and FAST). Vertical boxes represent the range observed between the
two replicates per treatment.



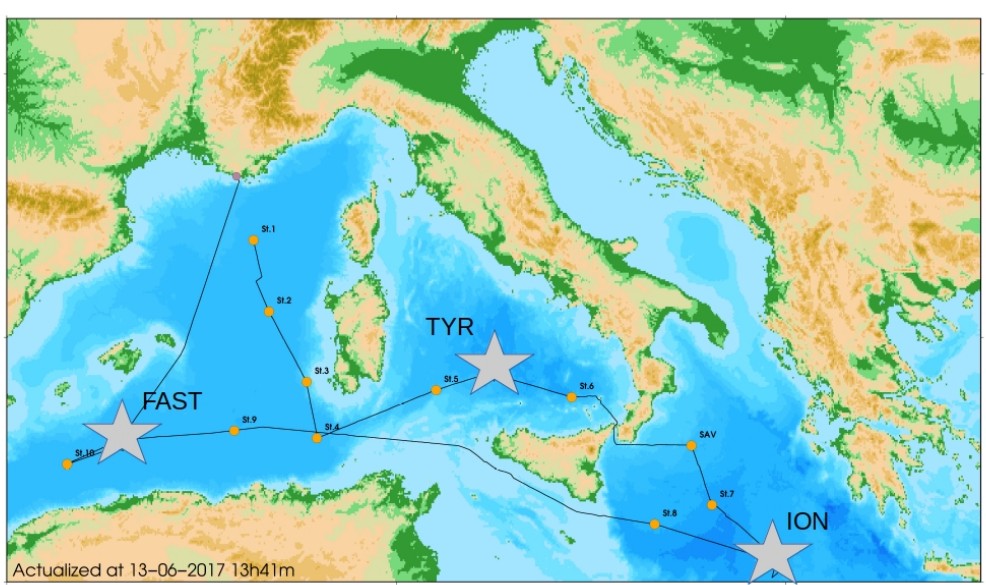


Fig. 1





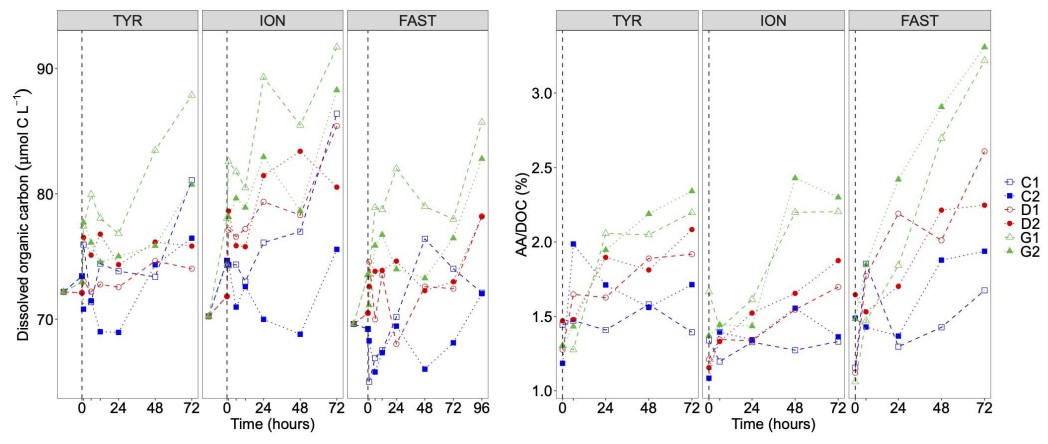


Fig. 2





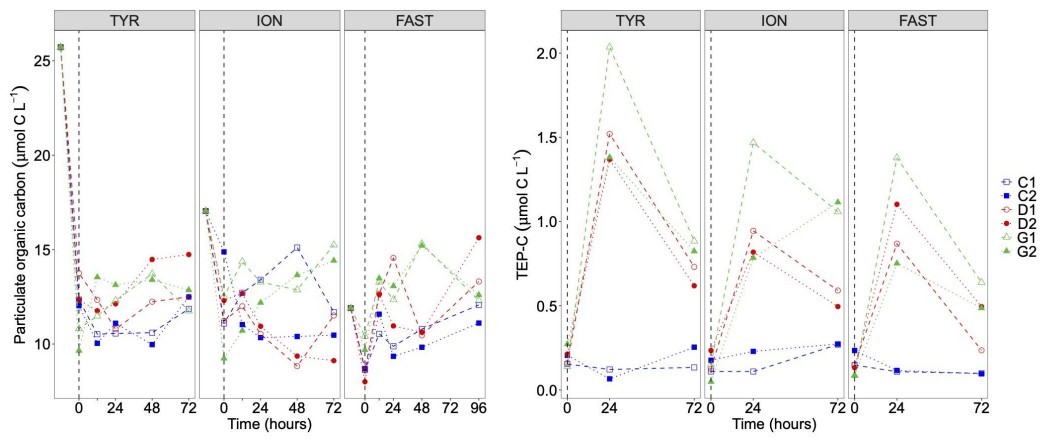


Fig. 3



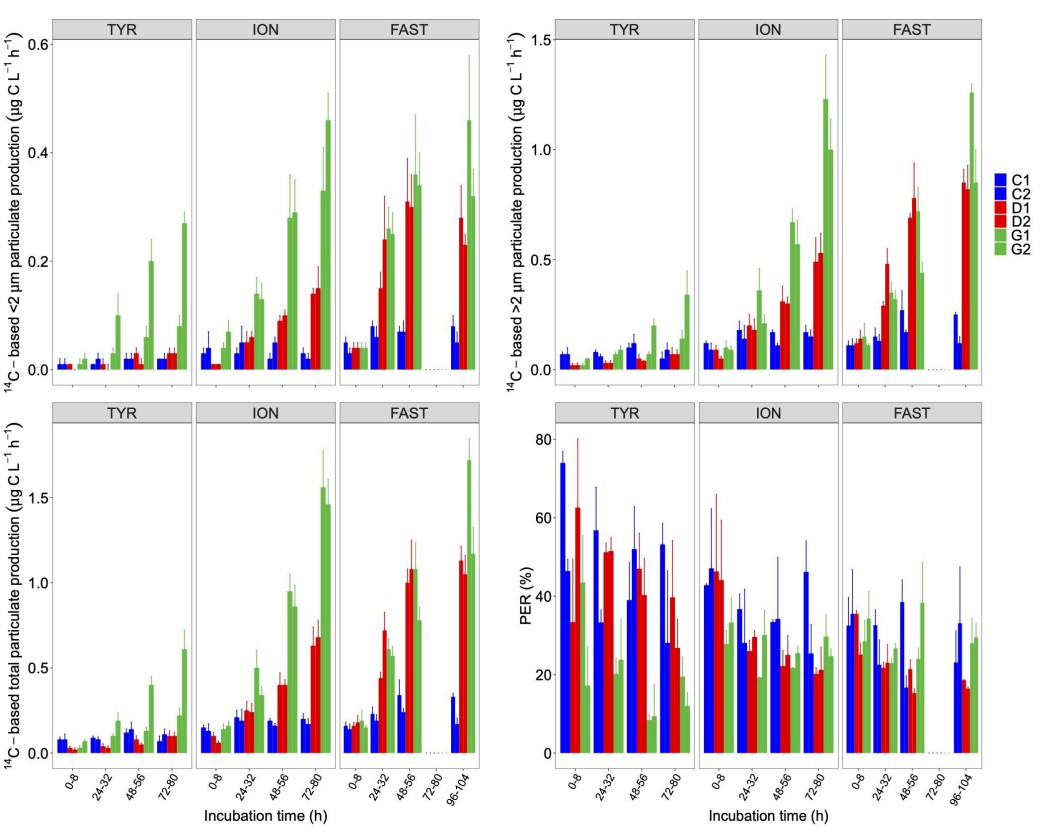


Fig. 4





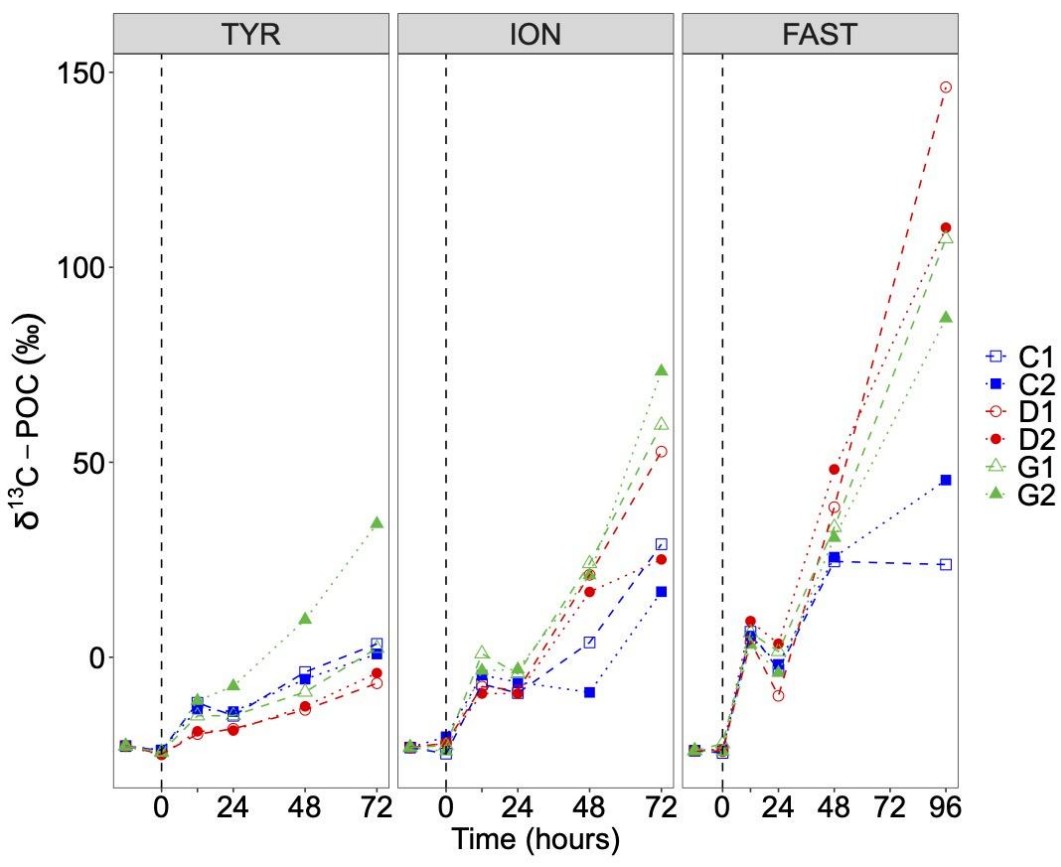


Fig. 5



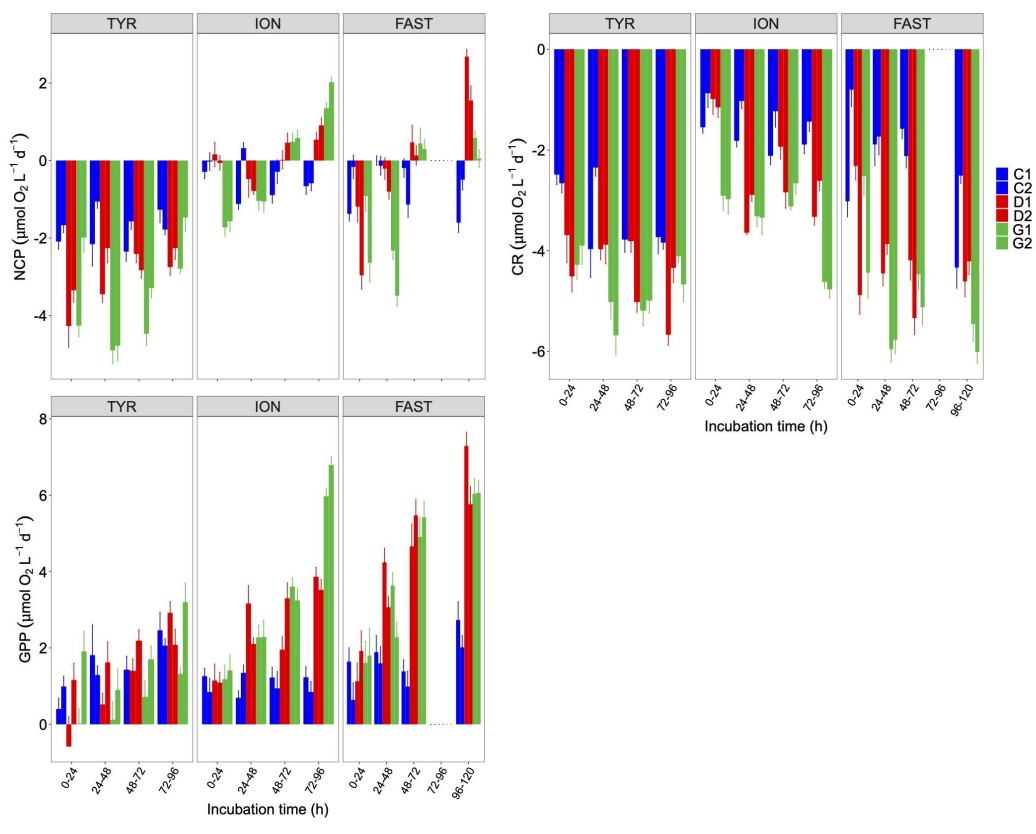


Fig. 6



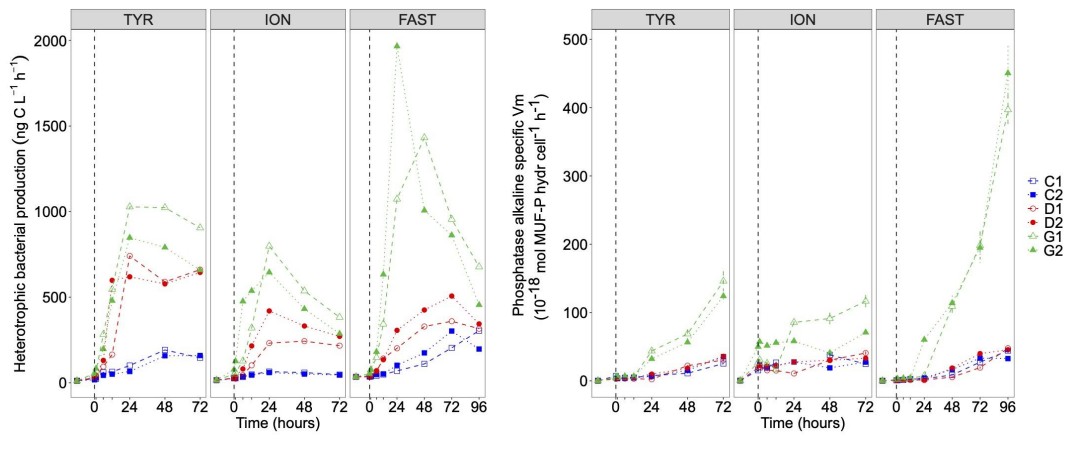


Fig. 7








Fig. 8





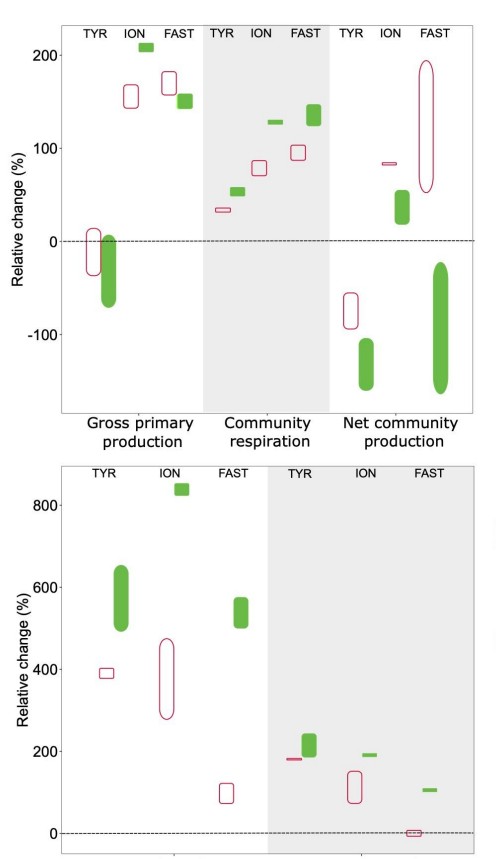

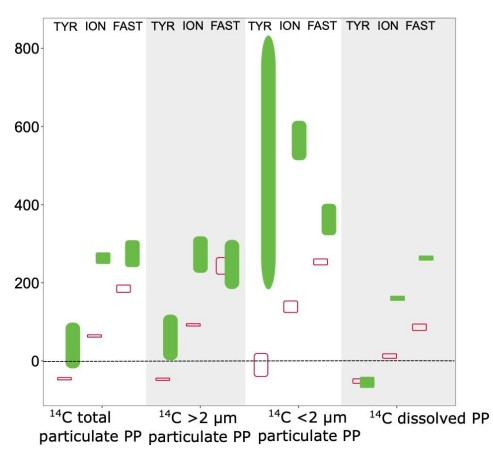

☐ Dust addition under current conditions
of temperature and pH

◼ Dust addition under conditions
of temperature and pH projected for 2100
ΔT = + 3°C and ΔpH = -0.3


Fig. 9