# Peer review of "Impact of dust addition on the metabolism of Mediterranean"

_Biogeosciences, 2021_

## Referee Comment (RC4)

**Review on: Impact of dust addition on the metabolism of Mediterranean plankton communities and carbon export under present and future conditions of pH and temperature**

**General comments**

The presented work provides valuable insights into the short-term response (within 72h or 96h) of the plankton community to dust input in oligotrophic low nutrient, low chlorophyll waters of the Mediterranean sea under present-day and future conditions. The authors show the time-evolution of the response leading to i) either a shift towards even stronger net-heterotrophy or, with a time lag of 2 days, ii) to a shift towards net-autotrophy of the plankton community.

In general, the authors present a rich dataset which was part of an even bigger effort, the PEACETIME project, to shed light on the role of dust input into Mediterranean waters.

The manuscript is well structured. However, I believe, the material could be further condensed and the readability and clarity increased. For example, often 'this' and 'that' are used. I suggest to be explicit to what you refer to make it easier for the reader. I tried to highlight some cases, where such clarification is needed in the specific comments - but please consider to go though the whole text.

I have two major concerns with the presented material:

1. While being aware about the enormous amount of work and the limited ship time to gain such dataset, the limited number of replicates of the experiments (using only duplicates instead of e.g. triplicates) makes it hard to draw statistically meaningful conclusions. This limits the value of the otherwise valuable experimental setup and study. I believe, this should be considered in potential follow up experiments.

2. The title promises to provide insights into the response of carbon export to dust input. First, a time evolution of export fluxes would have been certainly of value, which should be considered in a future application of the experimental design. Second, a few more information and discussion on the tank design and its (potential) effect on particulate fluxes would be helpful (e.g. is the energy input comparable to typical dust event situations, how does the circulation affect aggregation dynamics, do you account for ongoing grazing and remineralization in the 'sediment trap' of the tank, do you expect TEP to collect at the surface - away from your sampling valve - if so, in how far does it matter for export?). Third, I miss data and discussion on observables mentioned in the methods part, i.e. total carbon, lithogenic and biogenic silicates and calcium in the exported material. Further, I would have expected a stronger discussion on the aspect of export, e.g. also with regards to the ratio between POC and TEP-C production and export flux. It also seems as if there could be a relation between the community state and export flux.

Further, a discussion on questions related to the following points could be of value:

- In how far can be the present plankton community (and thus its response to dust input) regarded as adapted to future climate conditions (after a short period of adjustment time)?

- What are the potential consequences of the response within the studied time frame on longer time periods?

- What are the consequences of your findings on future modeling strategies?

Please find more specific comments below.

**Specific comments on the text**

p.2, l.42-44 A bit more explanation is needed on DOM as precursors for TEPs and thus aggregation of minerals - the connection between DOM and aggregation of dust particles is not immediately clear

p.3, l.48 What is 'This potential...' referring to? - be explicit.

p.4,l.56-57 ...(Longhurst et al. 1995). Although phytoplankton production in LNLC areas is limited ...

p.4,l.60 ...in LNLC regions and as such ...

p.5,l.85 ...Ridame and Guieu, 2020). However, no clear ...

p.6,l.108 Since you refer to biological remineralization processes being affected by ocean warming, write '...weaken the ocean biological $CO_2$ sink in the future ...' (to not confuse it with physical effects on $CO_2$ uptake).

p.6,l.117-120 ...trophic levels. Their study was conducted under nutrient-depleted conditions (Maugendre et al.., 2017b). Hence, there is still a need ...nutrient availability.

p.6,l.151 I have a number of questions regarding the setup of the tanks:

1. How efficient is sedimented material transported to the sediment trap? Or got material stuck to the tilted side walls and wasn't captured?

2. Was the sediment trap somehow poisoned to avoid remineralization and grazing on settled material? Or do you underestimate POC sedimentation fluxes?

3. How large (its area) was the propeller used, which direction of flow field was induced and how much energy was put in - how does the induced mixing rates compare to in situ conditions (also under wet dust input conditions)? And to phrase it broader: What is the propeller effect on the sedimentation flux?

p.9,l.169 Can you briefly mention the mean/median grain size?

p.12,l.239 ...regimes as in the ...

p.13,l.264 We followed the time evolution . . .

p.14,l.275 . . . frequency as for . . .

p.14,l.284 Ref. for standard needed

p.18,l.357 Which stoichiometry are you assuming to calculate the factor two? Or provide a reference.

p.20,l.387 I think, a brief description of the general environmental settings in which the experiments were carried out, would help, before going into the details of experimental results. Particularly such information of a pre-occured dust input event at TYR would aid the reader to understand the state of the plankton community.

p.21,l.422 I suspect you mean 'general positive trend' (as opposed to acceleration via an increasing trend)

p.21,l.423 Here and throughout the text: when speaking about variability, you seem to refer to differences between experiments and not to variability in the statistical sense as deviation from the mean. I suggest to move either to 'differences' or to define variability at the first occurrence as difference between the experiments.

p.22,l.446 positive trends (see above)

p.22,l.448-449 what are you referring to? Which final values and 3 % of what?

p.23,l.455 'this parameter' → 'DOC concentration' (if I am not mistaken, otherwise please fill with the right parameter name)

p.24,l.476-477 why is it an important discrepancy? - do you somewhere come back to this statement to explain it? In the figure, to my eye, I see two times the same green color, so it's not straight forward to see, which is G1 and G2 (I assume the order matters, but it could be clearer)

p.24,l.482-485 I am a bit confused here by increasing versus decreasing values. So the ratio between DOM and POM production shifted towards POM production and therefore the %PER was decreasing?! I guess, the sentence could be written in a clearer manner.

p.24,l.484 at this station → at station ION (please re-check)

p.24,l.486 at this station → at station ION (please re-check)

p.24,l.489 at this station → at station ION (please re-check)

p.24,l.495 Start with: In contrast to station ION, at station FAST was much less . . .

p.25,l.509 incorporation into

p.25,l.511 At station TYR,

p.25,l.512 under present-day environmental conditions

p.26,l.520-521 maybe add a gray line in FAST at 72h to make comparison easier

p.27,l.560 refer again to Tab. 2 at end of sentence.

p.28,l.571 instead of a continuous increase

p.28 Sec 2.4: I am missing information of how much of the dust was recovered from the 'sediment trap'. Additionally, you were mentioning the measurements of BSi etc. in the methods part and don't show it here (or anywhere else in the manuscript). I wonder why? See also my general comment.

p.28 No mentioning and summary of Fig.9 in the results part?

p.29,l.599-602 A rough back-of-the-envelope calculation for the flux induced by the loss of POC at TYR between t-12h and t0 (using an initial POC difference of about $15\,\mu\mathrm{mol\,C\,L^{-1}}$ from Fig. 3) under the assumption of a homogeneous water body:

$$\begin{aligned} F &= 280\,\mathrm{L} \cdot 15\,\mu\mathrm{mol\,C\,L^{-1}} \cdot \frac{1\,\mathrm{mol}}{10^6\,\mu\mathrm{mol}} \cdot 12\,\mathrm{g\,mol^{-1}} \cdot \frac{1000\,\mathrm{mg}}{1\,\mathrm{g}} \cdot \frac{1}{0.36\,\mathrm{m^2} \cdot 3\,\mathrm{d}} \\ &= 46.7\,\mathrm{mg\,C\,m^{-2}\,d^{-1}} \end{aligned}$$

provides a very different picture from what has been found in the 'sediment trap' of the control tanks, particularly, when also considering the stoichiometry for the POM/POC ratio (which would make a factor of 2 according to your methods part 2.3). This is already of the order of sedimentation fluxes or even higher than in the dust addition experiments. I am a bit concerned about the results related to the export of particulate matter. As already pointed out in the general comments part, a few more checks and a deeper discussion might help.

p.29,l.604-606 A higher TEP-C content is not really visible in Fig. 3. Given the higher POC content at t0 at TYR for C1/C2, you could potentially even come to the opposite conclusion (i.e. higher TEP-C/POC ratio at FAST).

p.30,l.629 Provided that dust input happened, it seems as if dust input frequency might play a role in determining the evolution of the plankton community. Is there anything know about it (frequency of events, expected changes in future, etc.)?

p.31,l.635 'optimal' or 'favorable conditions'? Looking at the dust input experiments, it seems to me that BP was not at its maximum.

p.31,l.642 at station FAST, as shown...

p.32,l.664 under present-day environmental conditions

p.32,l.670 here and throughout the manuscript, specify the main limiting nutrient - not everyone is familiar with the biogeochemistry of the Mediterranean sea

p.33,l.678,679,687 instead of 'this station' specify the station explicitly.

p.34,l.702-704 Please explain a bit more detailed.

p.35,l.723-725 Why is it higher under future conditions?

p.35,l.733 simply 'more in a steady state and less stressed' or maybe ' more in balance and less stressed'?

p.36,l.754 I suspect the reference needs to moved to '. . . new nutrients (e.g. . . . ) and ' - since the study of Moutin took place before your studies

p.37,l.777 more important → higher (or, in which sense more important?)

p.38,l.790 why don't you refer to the decreasing TEP-C as shown in Fig. 3?

p.38,l.799-804 The reference in line 801 cannot really refer to your experiments, so I would suggest to reformulate this part in terms of what Bressac and Guieu 2013 found and how that relates to your study.

p.39,l.822 this treatment → future conditions (or treatment G compared to . . . )

p.42,l.899 weakening of the $CO_2$. . .

**Specific comments on figures and tables**

Fig.4 I cannot distinguish between the individual tanks. I suspect the order matters. Since you refer to individual tanks in the text, it would be helpful to be able to distinguish between the tanks.

---

## Author Response (AR1)

**Co-editor-in-chief**

In light of your answer to the reviewer comments, I recommend publication with minor revisions, provided the answer to the comments are reflected in the new manuscript version. I would also like to point out that all reviewers commented on the length and large amount of data presented in the manuscript. I would, therefore, urge you, when possible, to focus on the relevant.

*We would like to thank the co-editor in chief for allowing us to submit a revised version of the manuscript. We agree with the referees and the co-editor in chief that our manuscript is lengthy, however we believe that all parameters/processes that are shown are important in order to have a full overview of what happens in the tanks during the three experiments. Unfortunately, the referees were not clear on what they believe could be reduced (or removed), and even suggested to add more discussion or to show more parameters.*

**Associate editor**

the answers to the referee's comments are complete and fulfilling. After all the modifications reported in the answers to the referees are transferred on the manuscript, it can be published on BG.

Regarding the question arises from Referee 3 regarding the table reporting the meaning of abbreviation, I think there is not need to this table as it is true that abbreviation are a lot, but many of them are commonly used in this research field.

*We would like to thank the associate editor for allowing us to submit a revised version of the manuscript.*

**Reviewer#1**

*We would like to thank Anonymous Referee #1 for her/his comments and suggestions on our manuscript. We acknowledge that the conclusion section of our submitted manuscript requires improvements. We will do our best to revise the manuscript accordingly. The suggestions for future work from the reviewer are very interesting and stimulating.*

Based on in situ experiments, this paper aims to quantify the impact of global change on the microbial food web and carbon export in an oligotrophic environment (Mediterranean Sea) during atmospheric dust deposition. The hypothesis underlying this work considers that an increase in temperature and a decrease in pH should reduce the intensity of the oceanic biological carbon sink (i.e. export production), especially in oligotrophic systems.

'Mesocosm' experiments have been undertaken in three different regions of the Mediterranean Sea (Ionian Sea, Tyrrhenian Sea and Algerian Basin) following an East-West gradient of production regimes. Each experiment is built on three distinct forcing conditions with small mesocosm duplicates, during 4 days: a reference (C), a realistic dust pulse (D), a realistic dust pulse together with a 3°C temperature increase and a 0.3 pH unit decrease. Numerous observations on stocks, fluxes, and physiological parameters are made during these incubations, and the discussion is based on these observations. A

companion paper presents the experimental design and discusses the results in terms of stocks, while this paper discusses fluxes and food web functioning.

The main message of the article is twofold: the impact of a dust disturbance depends drastically on the initial state of the microbial food web, and anthropogenic modifications (temperature, pH) only slightly modify the impact of this dust addition.

The article is interesting, even if the presentation of the numerous results (three experimental conditions in three different regions) can be difficult to follow. Indeed, due to the multiplicity of simulated situations and the rather small number of similar experiments (in terms of statistical robustness), the discussion is obviously complicated, with each experiment deserving specific comments. Nevertheless, within the limits of the proposed exercise, the two main conclusions of this work are robust.

There is not much to say about the data and the extensive and interesting discussion it prompts, very often based on realistic and specific hypotheses based on clues provided by the data, but which cannot be rigorously tested. Considering that this type of work is time consuming and quite expensive, the article could have ended with less "lazy" and general conclusions. In fact, the two main results of the article should have led to more directed conclusions. As an example, two sentences are rather vague, do not really come from the work presented, and do not really provide any new information.

"observations over longer temporal scales are probably required to ascertain the interactive effects  of these stressors in the coming decades": What types of observations and what time scales? Are the authors interested in long time series and how do they relate to this work, i.e. short-term mesocosm experiments? Very provocatively, are these experiments useful? How do you link the different time scales ?

*We fully agree that this conclusion is neither clear nor relevant. We propose to replace that sentence by:*

*"Although a longer experimental period would likely be necessary to clearly support an impact of future conditions on export, those changes occur on a long time scale that cannot be easily mimicked by experimental approaches. Only in situ co-located observations (atmospheric flux/export in sediment traps) over long temporal scales would be necessary to ascertain the interactive effects of these stressors at the decadal time scale."*

"As a consequence of a stronger sensitivity of heterotrophic prokaryotes to temperature and/or pH, the ongoing warming and acidification of the surface ocean will result in a decrease of the dust fertilization of phytoplankton in the coming decades and a weakening the $CO_2$ sequestration capacity of the surface oligotrophic ocean". This is the last sentence in the article. It does not really come from the results presented here, which in fact show the opposite (no change in exports).

*We do not fully agree with the reviewer here. Our results clearly show a decrease in net metabolism with warming/acidification. Although these results could not be linked to a decrease in export efficiency, most likely due to experimental bias, e.g. small fluxes*

*compared to those related to ballasting from dust, a decrease in net metabolism undoubtedly involves a decrease in the capacity of surface seawater to pump atmospheric CO2. The sentence was modified to make it clearer that we are referring to absorption of atmospheric CO2 by surface waters.*

On the other hand, the two main conclusions of the article should raise interesting questions for future work:

●       Important role of the initial state of the microbial food web: how to address this major issue and how to draw more general conclusions about the impact of the initial state of the ecosystem? What kind of experiments or in situ observations? Is it necessary and how to obtain the preconditions for the experiments or observations, the history of the food web? It is impossible to get the whole picture if this question about the initial state is not addressed (it is not possible to perform an infinite number of experiments, and a rational approach must be proposed). It will not be possible to address the question of the minimum amount of dust (nutrients) released to obtain a "positive" response from the microbial ecosystem (i.e., an increase in export) without addressing, according to the conclusion of the paper, in a more generic way the competition between autotrophs and bacteria for nutrients, and the link with temperature/metabolism. Besides theoretical considerations and possibly modeling approaches, what to do in terms of observations and experiments?

We agree that multiplying the number of experiments will not solve the issue, this is why we choose to perform the 'same' experiment in different waters. Doing this, our results could be translated into process parameterization made possible by the variety of tested waters, environmental stressors and responses. Such parameterization shall be used in biogeochemical models coupled to ocean dynamics that can depict the spatial and temporal impacts following a deposition in surface waters which biogeochemical properties is dependent on many factors (including successive dust depositions that can be tested in the model and degree of oligotrophy). This small paragraph will be added to the revised version.

●       Contrary to the expectations, the results of the experiments presented in the paper are not really convincing, as there is no real impact of global change on the export. If the basic hypothesis holds (e.g. there should have an impact in oligotrophic regimes), a different approach could be proposed, either in terms of experiments (number of tanks to get robust results, with the issue of feasibility, length of the experiments, types of observations) or in terms of sampling different regions and foodweb structures.

There is also the problem raised by multi-stress experiments. The combinatorial analysis makes the number of potential experiments impossible to address, if there are more than two/three stressors. Although the experiment with only a pH perturbation (or a temperature perturbation) was not undertaken, as the study was already quite cumbersome, the work seems to show that pH plays a minor role compared to temperature. Is there anything to be learned from this result (absolute necessity to do an experiment with only a pH perturbation, or to focus on temperature only, or on temperature and a possible additional stressor)?

*We do not fully agree with the reviewer but this shows that this paragraph was not clear and will be modified accordingly. As we did not discriminate pH from temperature effects (they occur together), we cannot conclude that our "work seems to show that pH plays a minor role compared to temperature". We still believe that a dedicated study is needed to discriminate temperature from pH effects although our working hypothesis would be that temperature exerts a much stronger control on community functioning than pH (as shown in our previous study from Maugendre et al. 2015). Our experimental system now comprises nine tanks and we could imagine in the near future running an experiment with duplicated tanks (triplicates for controls) and four conditions.*

The richness of the study results from the large number of parameters (stocks and flows) measured. Since the work is aimed at the export flux, a discussion based on eratio could provide a complementary perspective. As the experiments are obviously not steady state, since they are perturbation experiments, this may not be easy to do, but a balance (integrated in time during the duration of each experiment) of the different fluxes, i.e. network pathways, between the main boxes could be presented to get an overall view of the microbial functioning, and the differences between the three ecosystems.This could possibly be the objective of a complementary article.

*We agree. We initially had the objective to establish a budget and present it in this manuscript. However, we felt that it already contained quite a lot of information and we finally came to the same conclusion as the reviewer that this should be the objective of a complementary article.*

Last two remarks:

●      Line 301: "Gross primary production (GPP) was calculated as the difference between NCP and CR". It is the sum, and not the difference.

*As mentioned in the text : NCP and CR were estimated by regressing O2 values against time, and **CR was expressed as negative values**.*

   *Therefore, NCP = GPP + CR, and GPP = NCP - CR*

●      Figure 9: it seems that this figure is not consistent with Fig. 10 of the companion paper, at least for PP and BP.

*The reviewer is correct, however the two plots are not showing the same data. Fig. 10 of the companion paper is showing the "Maximal relative change (%)" while in this figure we show the "Relative difference (%) between integrated rates". We understand this might be confusing but we felt it was important in this second paper to show differences of the integrated rates instead of maximal differences.*

In conclusion, the data obtained during this cruise are very rich and valuable, despite the small number of replicated experiments, the discussion, although sometimes lengthy, is interesting, the conclusions are robust, and the paper should be published with small modifications (checks of Figure 9 against Figure 10 of the companion paper should be made, if there is indeed an inconsistency). Nevertheless, one may regret that the

conclusion is rather vague and agreed upon, although this work opens up real challenges and important scientific questions.

*Many thanks again. We will do our best to improve the conclusion section as suggested.*

**Reviewer#2**

*We would like to thank Anonymous Referee #2 for her/his comments and suggestions on our manuscript. On one hand, we acknowledge that the conclusion section of our submitted manuscript requires improvements. We will do our best to revise the manuscript accordingly. On the other hand, we do not really understand what is meant by the reviewer when arguing that: "Maybe it would be better to focus on some aspects of the processes described in order to achieve clarity and remove the impression of vagueness, both in the results and in the conclusions, which should be clearer and not indefinite, as they appear.". We understand that this manuscript contains quite a lot of data, however it seems very difficult to focus more on, for instance, phytoplankton metabolism without fully discussing the effects of dust/warming/acidification on bacterial activities. As we do not know exactly what was suggested, and as this issue was not raised by the other reviewers, we will keep all data as presented in the submitted version of the manuscript.*

First of all I wish to congratulate the authors for the huge quantity of data and measurements they have presented, which makes the article interesting and stimulating.

*Many thanks, much appreciated.*

Nonetheless some observations are to be made in order to improve the manuscript and make its reading and interpretation easier.
The data presented are maybe too many and the entire manuscript is too long and difficult to follow.
Maybe it would be better to focus on some aspects of the processes described in order to achieve clarity and remove the impression of vagueness, both in the results and in the conclusions, which should be clearer and not indefinite, as they appear.

*See general comment above.*

The paper often reminds to the other article (Gazeau at al. 2020) where the general situation of the stations studied is described, but this makes difficult for the reader to have a clear image of the situation, unless studying the other article. A very short description could be useful, especially for the phytoplanktonic community, and a short description of the different characteristics of the three stations, as the authors say the change in the community depends on their initial state.

*A brief paragraph will be added at the start of the discussion section to present the biogeochemical characteristics of the tested waters as discussed more in detail in Gazeau et al. (2020). However, as mentioned by several reviewers, the manuscript is already quite lengthy ("too long and difficult to follow" as written by the reviewer) and we would like to keep this section, which is fully detailed in the companion paper, as short as possible.*

Data show a general great variability, even between the replicates, as the authors themselves underline. Is there an explanation for this? In certain cases it is quite high and might invalidate the whole experiment.  If  there isn't a reason to keep the replicates separate, cannot you consider to average the two replicates? This can help in the understanding of the figures, especially fig. 2, 3, 4, 6, 7.

Axis title should be enlarged to simplify the reading.

*We acknowledge that variability is, for some parameters and experiments, at least visible. The choice for a duplicate approach is fully discussed in the companion paper (unfortunately still not readable in its final revised version…); section 4.2. Critical assessment of the experimental system and methodology: "The relatively low number of experimental units that could be installed inside an embarkable clean container restrained our possibility to consider more than two replicates per treatment. Fortunately, as already said, differences between duplicates were, for the vast majority of studied variables and processes, lower than differences between treatments and appear acceptable considering the difficulty to incubate plankton communities for which slight differences in their initial composition can translate into very important differences in dynamics (Eggers et al., 2014)."*

*Unfortunately, duplicates do not allow averaging and even less calculating standard deviations which will lead to an obvious mathematical error and a loss of information (we do not want to hide this variability).*

*Larger figures will be provided in the revised version in order to improve visibility.*

The projection of what the dust impact could be in the future is very interesting, but clearer conclusions should be addressed.

*We agree with this assessment that is shared by the other referees and will clarify and extend the conclusion section in the revised version of the manuscript (see reply to RC1 for more details).*

Finally, the paper presents data of current interest, and deserves a clearer presentation, so it is worthy publication in Biogeoscience with the improvement suggested.

*Many thanks again for your positive comments.*

*We would like to thank Anonymous Referee #3 for her/his comments and suggestions on our manuscript.*

In this manuscript, Gazeau and colleagues present a significant work about metabolism adaptation of Mediterranean phytoplankton to dust fertilization, under present conditions and in an acidified future ocean enriched in CO2, with a substantial very new data set. They also provide new insights about particle export in this context, that are a precious contribution for modeling. The paper is well organized, but quite long with a lot of concepts that make it a little bit hard to follow. It does not help to clear the main ideas and conclusions easily. The paper is recommended for publication, also minor changes should be considered before this, that are listed below.

First, a table of the abbreviations would be very helpful.

*This is not a problem to add such a table. However, we do not think our manuscript contains too many abbreviations to justify a table. We will rely on the editor's opinion for that matter.*

L57 and 59: more recent references are needed

*Although we think that those references are accurate and pertinent, we added other references. "Low Nutrient Low Chlorophyll (LNLC) areas represent 60% of the global ocean surface area (Longhurst et al., 1995; McClain et al., 2004). Although phytoplankton production in these areas is limited by the availability of nitrogen, phosphorus and iron, it accounts for 50% of global carbon export (Emerson et al., 1997; Roshan and DeVries, 2017)."*

L59: same sentence as in the abstract. Not a problem but not very elegant.

*Sentence has been modified: "Atmospheric dust fluxes represent a significant source of these nutrients to surface waters in LNLC regions and as such could play a significant role in stimulating primary production".*

L76: if I understand correctly the metabolic balance is not enough to draw conclusions about the biological pump, the NCP can only provide information about surface water, not about what happen at greater depth, therefore you can't really constrain the efficiency of the export from this only.

*The reviewer is correct and the sentence has been modified to : "The metabolic balance (or net community production, NCP) is defined as the difference between gross primary production (GPP) of autotrophic organisms and community respiration (CR) of both autotrophic and heterotrophic organisms, revealing the capacity of surface waters to absorb atmospheric $CO_2$."*

L80: Please add more general references. This is two very local studies.

We replaced these references by a recent review paper (Desboeufs, 2022) dealing with atmospheric inputs at the scale of the whole Mediterranean Sea.

L136: it would be great to have a quick summary of the paper

We could add this sentence : "A companion paper presents the general setup of the experiments and the impacts of dust under present and future environmental conditions on nutrients and biological stocks (Gazeau et al., 2021). In this paper, we show that the effects of dust deposition on biological stocks were highly different between the three investigated stations and could not be attributed to differences in their degree of oligotrophy but rather to the initial metabolic state of the community. We further demonstrated that ocean acidification and warming did not drastically modify the composition of the autotrophic assemblage with all groups positively impacted by warming and acidification.

L158: when were the last dust events in the three areas?

As discussed further in the paper (L628 and section concerning impact at TYR), only the TYR station encountered a dust deposition prior to the cruise. For the other two sites, there was no evidence from dust forecasts (observations and models: see PEACETIME Operation Center (http://poc.sedoo.fr/)) that an event occurred at least three weeks before the sampling.

L168: please do a quick summary of the experiment: why these concentrations, what is the composition of the dust? Does it reflect a pure lithogenic input, does it have an anthropogenic component?

This will be added in this paragraph: "Briefly, the fine fraction (< 20 µm) of Saharan soils collected in southern Tunisia, which is a major source of dust deposition over the northwestern Mediterranean basin, was used in the seeding experiments. The particle size distribution showed that 99% of particles had a size smaller than 0.1 µm, and that particles were mostly made of quartz (40%), calcite (30%) and clay (25%; Desboeufs et al., 2014). This collected dust underwent an artificial chemical aging process by addition of nitric and sulfuric acid ($HNO_3$ and $H_2SO_4$, respectively) to mimic cloud processes during atmospheric transport of aerosol with anthropogenic acid gases (Guieu et al., 2010a, and references therein). To mimic a wet flux event of 10 g m-2, 3.6 g of this analog dust were quickly diluted into 2 L of ultrahigh-purity water (UHP water; 18.2 MΩ cm−1 resistivity), and sprayed at the surface of the tanks using an all-plastic garden sprayer (duration = 30 min). The intensity of this simulated wet deposition event (i.e. 10 g m-2) represents a high but realistic scenario, as several studies reported even higher short wet deposition events in this area of the Mediterranean Sea (Bonnet and Guieu, 2006; Loÿe-Pilot and Martin, 1996; Ternon et al., 2010)."

L201: what is the percent of agreement between the replicates? Did you do some blanks?

L213: Same as above for TCHO, percentage of agreement between replicates and blanks?

*This was added to the section. For TCHO, the variation coefficient between duplicate measurements was 7% on average. For TAA, the variation coefficient between duplicate measurements was 8% on average. For TCHO and TAA, instrument blanks were performed with MilliQ water. The detection limit was calculated as 3x the blank value, which is ~1 nmol L-1 for both parameters.*

L215: how did you choose the different times of sampling for the different parameters and following which criteria did you chose to end the experiment?

*Several aspects had to be considered when defining the sampling times for each parameter: (1) the amount of water needed, as we did not want to have a final (before the last sampling) volume in the tanks of less than 50% of the initial volume, (2) the time needed to process the samples (or perform incubations for processes), and (3) obviously the analytical costs (both human and pecuniary).*
*Regarding the duration of the experiments, we have added this in the revised version of the companion paper. From Gazeau et al. (2021, BG): "The experiment at stations TYR and ION lasted 72 h (3 days) whereas the last experiment at station FAST was extended to four days. This relatively short duration of the experiments was constrained by the time available between stations and the time needed to properly clean the tanks between the experiments, following the protocol described by Bressac and Guieu (2013). As a larger time window was possible at the end of the cruise, the experiment at FAST was extended to four days."*

L225: Does the measures of the two filters agree? Did you do several measures on the same filter? What does the blank represent compared to the samples?

*For TEP, the coefficients of variation averaged 28%. All TEP values have been blank corrected. Blanks were always <1% of sample values. This was added to the section.*

L239: "compared to" instead of "then" I think

*Modified to, following R#4 suggestion: "Bottles were incubated for 8 h in two extra 300 L tanks maintained under similar light and temperature regimes as in the experimental tanks"*

L254: did you do some blanks and replicates? What is the standard deviation associated to the measurement?

*Yes, we performed triplicate measurements in the light and one in the dark. From the submitted version: "From each tank, four polystyrene bottles (70 mL; three light and one dark bottles) were filled with sampled seawater and amended with 40 µCi of NaH14CO3.". Standard deviations are shown in Fig. 4.*

L282: same as the precedent comment

*Due to the amount of volume necessary (2 L), no replicated sampling could be done. Corrections were made from blank measurements that were performed on pre-filtered seawater from the tanks.*

L301: according to your first definition it's an addition not a difference

*We disagree. As mentioned in the text : NCP and CR were estimated by regressing O2 values against time, and **CR was expressed as negative values**.*
*Therefore, NCP = GPP + CR, and GPP = NCP - CR*

L343: Have you measured Na to check if the salt was correctly removed and does not contribute significantly to the weight?

*To remove the salt, the JGOFs protocol was followed (Knap et al., 1996). That protocol has been used routinely by the Service National de la Cellule Pièges for more than 30 years. The protocol has been established to remove all Na from seawater.*

L345 to 356: please quantify the blanks, agreement between replicates and the standard deviation

*The blanks were 1.1 % of the average concentration of the sample (thus negligible), replicates agreement was on average 0.3 - 2.3 % (and no standard deviation as we analyse only 2 aliquots over the 3 when the 2 first measurements agree (<5% difference) and that was the case for all the samples analysed in this study.*

L356 to 358: please provide the references those ratios come from

*Klass and Archer, 2002*
*Klaas, C., & Archer, D. E. (2002). Association of sinking organic matter with various types of mineral ballast in the deep sea: Implications for the rain ratio. Global Biogeochemical Cycles, 16(4), 63-1.*

L598: it would help to have a graph of comparison in the supplementary

*We do not think that having a plot containing 3 points would be very useful.*

L631: it would be better to do a citation, it is well-known that dust events provide Al

*We believe that giving the in situ evidence that a large dust deposition occurred a few days before the TYR station was occupied is more pertinent. The citation is now: Bressac et al., in rev. 2021 (Bressac, M., Wagener, T., Leblond, N., Tovar-Sánchez, A., Ridame, C., Albani, S., Guasco, S., Dufour, A., Jacquet, S., Dulac, F., Desboeufs, K., and Guieu, C.: Subsurface iron accumulation and rapid aluminium removal in the Mediterranean following*

*African dust deposition, Biogeosciences Discuss. [preprint], https://doi.org/10.5194/bg-2021-87, in review, 2021.)*

L808: maybe add a quick summary

*Agreed. "Gazeau et al. (2020) have already discussed temperature/pH mediated changes in nutrient uptake rates and autotrophic community composition in these experiments. Briefly, they showed that warming and acidification did not have any detectable impact on the release of nutrients from atmospheric particles. Furthermore, these external drivers did not drastically modify the composition of the autotrophic assemblage with all groups benefiting from warmer and acidified conditions. Here, we showed that the difference in the response of plankton community metabolism to dust addition under present and future conditions of temperature and pH was highly dependent on the sampling station (Fig. 9).".*

L784: you can cite studies on the ballast effect, of P. Lam for example

*We added a sentence later in the same section to refer to some key papers about ballast effect following RC4 comments.*

L894: you could highlight better how useful your work can be for modeling

*We added a short paragraph at the end of the conclusion about the possible link with models (see reply to RC1).*

L1282: Please add the dates of the cruise in the caption, the latitude and longitude on the map, the color bar for the bathymetry you represent. Enlarge the numbers of the station, as your study is part of a larger work it will help to compare with other publications.

*We actually have changed the figure to be consistent with the one shown in the companion paper. Latitude and longitude are now displayed. The dates of the cruise are not displayed in order to be consistent with the companion paper.*

The other figures are tiny and hard to read, except figures 5 and 8. Please enlarge the titles of the axis, especially the time. When you have several panels (in figure 4 or 6 for example), it would be better with letters for the different panels.

*Unfortunately, we cannot increase label sizes as otherwise labels from the different panels would overlap. However, we strongly believe that the fact that figures were tiny in the submitted version of the manuscript is linked to the portrait setup of the pages, this will be modified in the revised version.*

If not included in the points and if possible include the error bars on figures 2,3,5,7,8.

*We cannot provide error bars for parameters for which only two measurements were done. This is the case for DOC and AA (Fig. 2) and for TEP (Fig. 3). No replicated analyses were done for POC (Fig. 3), 13C-POC (Fig. 5) and for export (Fig. 8). We have added error bars*

*to Fig. 7, thanks for detecting the oversight. Please note that for parameters which were analysed in duplicates, the correspondence between these duplicates has been added in the Material and Methods following the reviewers' advice.*

*We would like to thank Anonymous Referee #4 for her/his comments and suggestions on our manuscript. We agree with most comments and modified/updated the manuscript accordingly. Below is a point-by-point reply. Briefly, the reviewer would like to see more discussion on the export. We have tried to clarify some points related to the protocol used to recover the traps and we now give more details on the factions that were exported. Nevertheless, we do not want in this manuscript to go into much detail on the potential relationships between TEPs and organic matter export, as this manuscript contains already a lot of data (as highlighted by several reviewers) and we are currently working on a manuscript fully dedicated to this. We would like to thank the reviewer for highlighting a very important mistake in Fig. 3. t-12h data were wrong for all three stations, we believe this comes from a wrong treatment of the dataset to draw the plot (copy-paste). The data shown are now coherent with values indicated in Table 1. We apologize for this mistake.*

General comments

The presented work provides valuable insights into the short-term response (within 72h or 96h) of the plankton community to dust input in oligotrophic low nutrient, low chlorophyll waters of the Mediterranean sea under present-day and future conditions. The authors show the time-evolution of the response leading to i) either a shift towards even stronger net-heterotrophy or, with a time lag of 2 days, ii) to a shift towards net-autotrophy of the plankton community.

In general, the authors present a rich dataset which was part of an even bigger effort, the PEACETIME project, to shed light on the role of dust input into Mediterranean waters. The manuscript is well structured. However, I believe, the material could be further condensed and the readability and clarity increased. For example, often 'this' and 'that' are used. I suggest to be explicit to what you refer to make it easier for the reader. I tried to highlight some cases, where such clarification is needed in the specific comments - but please consider to go through the whole text.

I have two major concerns with the presented material:

1. While being aware about the enormous amount of work and the limited ship time to gain such dataset, the limited number of replicates of the experiments (using only duplicates instead of e.g. triplicates) makes it hard to draw statistically meaningful conclusions. This limits the value of the otherwise valuable experimental setup and study. I believe, this should be considered in potential follow up experiments.

*We obviously agree with the reviewer. At that time, our experimental system comprised 6 tanks and did not allow considering triplicates. We have recently developed a new clean container which contains 9 tanks and will allow considering 3 treatments in triplicates.*

2. The title promises to provide insights into the response of carbon export to dust input.

First, a time evolution of export fluxes would have been certainly of value, which should be considered in a future application of the experimental design.

*Our experimental system allows for an easy collecting of sediment traps which could be done on a daily basis. This was not considered for this study as we expected the fluxes to be too low to be precisely quantified on a daily basis. We definitely agree with the reviewer that this should be considered for a future study.*

Second, a few more information and discussion on the tank design and its (potential) effect on particulate fluxes would be helpful (e.g. is the energy input comparable to typical dust event situations, how does the circulation affect aggregation dynamics, do you account for ongoing grazing and remineralization in the 'sediment trap' of the tank, do you expect TEP to collect at the surface - away from your sampling valve - if so, in how far does it matter for export?).

Third, I miss data and discussion on observables mentioned in the methods part, i.e. total carbon, lithogenic and biogenic silicates and calcium in the exported material.

*As indicated in section 2.3.5, those analyses were used to quantify the different fractions (Ca was used to determine CaCO3, a fraction that was also determined by difference between (total C and POC), similar concentrations were found. Lithogenic fraction was determined using LSi and the same proportion of lithogenic was found when calculated as (total mass - organic matter + CaCO3 + opal).*
*We added to figure 8 the distinction between the different fractions?*
*(see also detailed response concerning contrasted efficiency dust export in another comment below and added text).*

Further, I would have expected a stronger discussion on the aspect of export, e.g. also with regards to the ratio between POC and TEP-C production and export flux. It also seems as if there could be a relation between the community state and export flux.

*We agree with the reviewer that this would be interesting to link production of TEPs to export flux. We did not want to go into too much details in this manuscript that is already quite long and contains lots of data. We are currently working on a manuscript (Guieu et al., in prep.) that gathers all data acquired from dust additions experiments including those during the PEACETIME cruise, and in situ observations to better understand POC export following dust deposition in the ocean. In particular, we focus on the lithogenic carbon pump that is responsible for a significant POC export in all studied cases and strongly linked to TEPs production.*

Specific comments on the text

p.2, l.42-44 - A bit more explanation is needed on DOM as precursors for TEPs and thus aggregation of minerals - the connection between DOM and aggregation of dust particles is not immediately clear

We added this sentence to the abstract: "At ION and FAST, the efficiency of organic matter export due to mineral/organic aggregation processes was lower than at TYR and likely related to a lower quantity/age of dissolved organic matter present at the time of the seeding and a smaller production of DOM following dust addition. This was also reflected by lower initial concentrations in transparent exopolymer particles (TEP) and a lower increase in TEP concentrations following the dust addition, as compared to TYR."

p.3, l.48 - What is 'This potential. . . ' referring to? - be explicit.

Corrected to: "this impact"

p.4,l.56-57 - . . . (Longhurst et al. 1995). Although phytoplankton production in LNLC areas is limited . . .

Corrected to: "Low Nutrient Low Chlorophyll (LNLC) areas represent 60% of the global ocean surface area (Longhurst et al., 1995). Although phytoplankton production in these areas is limited by the availability of nitrogen, phosphorus and iron, it accounts for 50% of global carbon export (Emerson et al., 1997)."

p.4,l.60 - ...in LNLC regions and as such...

Corrected

p.5,l.85 -  . . . Ridame and Guieu, 2020). However, no clear . . .

Corrected

p.6,l.108 - Since you refer to biological remineralization processes being affected by ocean warming, write '. . . weaken the ocean biological CO2 sink in the future . . . ' (to not confuse it with physical effects on CO2 uptake).

Added

p.6,l.117-120 - . . . trophic levels. Their study was conducted under nutrient-depleted conditions (Maugendre et al.., 2017b). Hence, there is still a need . . . nutrient availability.

Corrected

p.6,l.151 - I have a number of questions regarding the setup of the tanks:

1. How efficient is sedimented material transported to the sediment trap? Or got material stuck to the tilted side walls and wasn't captured?

*We made sure that no material remained in the tanks while collecting the sediment collection bottles. In case some material was observed, by tapping the walls from the outside, this procedure was enough to collect all the material in the traps.*

2. Was the sediment trap somehow poisoned to avoid remineralization and grazing on settled material? Or do you underestimate POC sedimentation fluxes?

*The sediment traps were not poisoned, we felt it was way too risky in case of leakage for the communities.*

3. How large (its area) was the propeller used, which direction of flow field was induced and how much energy was put in - how does the induced mixing rates compare to in situ conditions (also under wet dust input conditions)? And to phrase it broader: What is the propeller effect on the sedimentation flux?

*On board R/Vs, the main turbulence comes from the movement of the boat and the propellers may only have a slight effect compared to in situ dynamics. Those systems are also used for on-land experiments and the exact same gentle turbulence effect being applied to all the tanks, allows for the comparison of similar turbulence conditions in the different treatments. We agree that, for in land experiments, some tests should be done to properly quantify the induced turbulence and the impact on the processes including the export.*

p.9,l.169 - Can you briefly mention the mean/median grain size?

*Following R#3 advice, we have added a paragraph describing the dust composition in the Material and Methods.*

p.12,l.239 - ...regimes as in the …

Corrected

p.13,l.264 - We followed the time evolution . . .

Corrected

p.14,l.275 - . . . frequency as for . . .

Corrected

p.14,l.284 - Ref. for standard needed

The standard used was caffeine (IAEA-600; https://nucleus.iaea.org/sites/ReferenceMaterials/Pages/IAEA-600.aspx), this was added to the text.

p.18,l.357 - Which stoichiometry are you assuming to calculate the factor two? Or provide a reference.

Klaas and Archer, 2002 added in the text
Klaas, C., & Archer, D. E. (2002). Association of sinking organic matter with various types of mineral ballast in the deep sea: Implications for the rain ratio. Global Biogeochemical Cycles, 16(4), 63-1.

p.20,l.387 - I think, a brief description of the general environmental settings in which the experiments were carried out, would help, before going into the details of experimental results. Particularly such information of a pre-occured dust input event at TYR would aid the reader to understand the state of the plankton community.

*Added*

p.21.l.422 - I suspect you mean 'general positive trend' (as opposed to acceleration via an increasing trend)

Absolutely, corrected to positive trend

p.21,l.423 - Here and throughout the text: when speaking about variability, you seem to refer to differences between experiments and not to variability in the statistical sense as deviation from the mean. I suggest to move either to 'differences' or to define variability at the first occurrence as difference between the experiments.

The term variability in our text always refers to differences between duplicates (and not experiments), therefore we believe we can use the term variability.

p.22,l.446 - positive trends (see above)

Corrected

p.22,l.448-449 - what are you referring to? Which final values and 3 % of what?

*Modified to: "The strongest increase was observed at station FAST in tanks G where final TAA/DOC ratios were above 3%."*

p.23,l.455 - 'this parameter' → 'DOC concentration' (if I am not mistaken, otherwise please fill with the right parameter name)

*No. "After dust seeding, POC concentrations did not show clear temporal trends for the three experiments although a slight general increase could be observed at station FAST. Furthermore, no impact of dust seeding and warming/acidification could be observed for this parameter." We do not see what is unclear with these sentences, however we could change to: "Furthermore, no impact of dust seeding and warming/acidification could be observed on POC dynamics".*

p.24,l.476-477 - why is it an important discrepancy? - do you somewhere come back to this statement to explain it? In the figure, to my eye, I see two times the same green color, so it's not straight forward to see, which is G1 and G2 (I assume the order matters, but it could be clearer)

*Modified to large discrepancy. The figures with bar plots were modified in order to easily discriminate between duplicates.*

p.24,l.482-485 - I am a bit confused here by increasing versus decreasing values. So the ratio between DOM and POM production shifted towards POM production and therefore the %PER was decreasing?! I guess, the sentence could be written in a clearer manner.

*The initial sentence was: "Although being also positively impacted and increasing with time, dissolved production appeared less sensitive than particulate production leading to an overall decrease of %PER at station ION following dust addition."*

*We propose to reformulate to: "At ION, both particulate and dissolved production increased following dust addition. As this positive impact was stronger for particulate than for dissolved production, this led to an overall decrease of %PER following dust addition."*

p.24,l.484 - at this station → at station ION (please re-check)

*Indeed, modified*

p.24,l.486 - at this station → at station ION (please re-check)

*No need to repeat, specified in the sentence just before.*

p.24,l.489 - at this station → at station ION (please re-check)

*Indeed, modified*

p.24,l.495 - Start with: In contrast to station ION, at station FAST was much less . . .

*Modified to: "However, in contrast to station ION, there was much less impact of warming/acidification on all measured rates at station FAST although ..."*

p.25,l.509 - incorporation into

*Modified*

p.25,l.511 - At station TYR,

*Modified*

p.25,l.512 - under present-day environmental conditions

*Not modified, we refer to present vs future environmental conditions throughout the text, we do not see why using a different terminology here.*

p.26,l.520-521 - maybe add a gray line in FAST at 72h to make comparison easier

*No data are shown at 72 h, we believe it is clear that no measurements have been done for this day at FAST.*

p.27,l.560 - refer again to Tab. 2 at end of sentence.

*Done*

p.28,l.571 - instead of a continuous increase

*Modified*

p.28 - Sec 2.4: I am missing information of how much of the dust was recovered from the 'sediment trap'. Additionally, you were mentioning the measurements of BSi etc. in the methods part and don't show it here (or anywhere else in the manuscript). I wonder why? See also my general comment.

*We agree that important information is missing. In the results section, the following sentences were added and the different fractions are now presented in Fig. S5.*

*"Only less than 30% of the dust introduced at the surface of the tanks were recovered at the end of the experiment (3 or 4 days after) in the sediment traps with TYR>ION>FAST. The composition of the exported material was quite similar for each experiment with no significant difference between D and G treatment with: 3-5% Opal, 4% organic matter, 35-36% CaCO₃ and 48-54% Lithogenic (Fig. S5).*

*and after L793:*

*"The recovery of the introduced dust (traced by the lithogenic mass recovered in the traps) was low (27% at TYR, ~20% at ION and 13-19% at FAST) reflecting that a majority of the dust particles (the smaller ones that are the most abundant according to the particle size distribution of the dust) still remained in the tanks after 3 or 4 days following dust addition. This has been already observed in pelagic mesocosms (Bressac et al., 2012) as those small particles can aggregate to organic matter and eventually sink. The higher export efficiency observed (TYR>ION>FAST) is likely linked to the higher initial abundance and higher production of TEPs during the experiment (Fig. 3).*

*Ref: Bressac, M., Guieu, C., Doxaran, D., Bourrin, F., Obolensky, G., & Grisoni, J. M. (2012). A mesocosm experiment coupled with optical measurements to assess the fate*

*and sinking of atmospheric particles in clear oligotrophic waters. Geo-Marine Letters, 32(2), 153-164.*

p.28 - No mentioning and summary of Fig.9 in the results part?

*No, this synthesis figure is presented in the discussion section, we believe this is the right place to show it.*

p.29,l.599-602 - A rough back-of-the-envelope calculation for the flux induced by the loss of POC at TYR between t-12h and t0 (using an initial POC difference of about 15 µmol C L−1 from Fig. 3) under the assumption of a homogeneous water body:
F = 280L·15µmolCL−1 · 1mol ·12gmol−1 · 1000mg · 1 106µmol 1g 0.36m2 ·3d
= 46.7 mg C m−2 d−1
provides a very different picture from what has been found in the 'sediment trap' of the control tanks, particularly, when also considering the stoichiometry for the POM/POC ratio (which would make a factor of 2 according to your methods part 2.3). This is already of the order of sedimentation fluxes or even higher than in the dust addition experiments. I am a bit concerned about the results related to the export of particulate matter. As already pointed out in the general comments part, a few more checks and a deeper discussion might help.

*We would like to thank the reviewer for detecting a very important mistake in our Fig. 3. We do not know what is the cause of this (bad copy-paste or else) but indeed T-12h values were completely wrong and not coherent with the values we indicated in Table 1. This has been fixed.*

p.29,l.604-606 - A higher TEP-C content is not really visible in Fig. 3. Given the higher POC content at t0 at TYR for C1/C2, you could potentially even come to the opposite conclusion (i.e. higher TEP-C/POC ratio at FAST).

*TEP-C was on average (all tanks as it was before dust seeding) 2.29, 1.69 and 1.67 micromol C/L at TYR, ION and FAST respectively leading to an overall TEP-C/POC ratio of 0.20, 0.14 and 0.19. We agree that, although TEP-C concentrations were higher at TYR, the "larger contribution to POC" as mentioned in our submitted version is not obvious. We then removed the end of the sentence.*
*"At t0, larger and more abundant TEP were measured at station TYR compared to the two other stations (data not shown)."*

p.30,l.629 - Provided that dust input happened, it seems as if dust input frequency might play a role in determining the evolution of the plankton community. Is there anything known about it (frequency of events, expected changes in future, etc.)?

*We wrote in the introduction that for the Mediterranean specifically: "dust deposition could increase in the future due to desertification (Moulin and Chiapello, 2006), although so far*

*the trend for deposition remains uncertain because the drying of the Mediterranean basin might also induce less wet deposition over the basin (Laurent et al., 2021)."*

*Although it is not clear whether there will be more deposition or not in the Mediterranean, this paper (Kok, J. F., Ward, D. S., Mahowald, N. M., and Evan, A. T.: Global and regional importance of the direct dust-climate feedback, Nat Commun, 9, 241, https://doi.org/10.1038/s41467-017-02620-y, 2018.) provides estimates of future dust loading on the global scale and regionally, including the Sahara.*

p.31,l.635 - 'optimal' or 'favorable conditions'? Looking at the dust input experiments, it seems to me that BP was not at its maximum.

*Agreed, favorable is a better term to use here.*

p.31,l.642 - at station FAST, as shown…

*Corrected*

p.32,l.664 - under present-day environmental conditions

*We prefer using present and future throughout the text*

p.32,l.670 - here and throughout the manuscript, specify the main limiting nutrient - not everyone is familiar with the biogeochemistry of the Mediterranean sea

*Done*

p.33,l.678,679,687 - instead of 'this station' specify the station explicitly.

*All corrected*

p.34,l.702-704 - Please explain a bit more detailed.

*We do not want here to reproduce what is written in the companion paper from Dinasquet et al. (2021). This manuscript was not available to the reviewer but is now online as a discussion paper (under review).*
*Dinasquet, J., Bigeard, E., Gazeau, F., Azam, F., Guieu, C., Marañón, E., Ridame, C., Van Wambeke, F., Obernosterer, I., and Baudoux, A.-C.: Impact of dust addition on the microbial food web under present and future conditions of pH and temperature, Biogeosciences Discuss. [preprint], https://doi.org/10.5194/bg-2021-143, in review, 2021.*

p.35,l.723-725 - Why is it higher under future conditions?

*We do not discuss future conditions in this section. As discussed in the next section, warming affects organisms by enhancing their metabolic rates.*

p.35,l.733 - simply 'more in a steady state and less stressed' or maybe ' more in balance and less stressed'?

*Corrected: "It seems that heterotrophic bacteria and phytoplankton were more in balance and less stressed at the start of the experiment at FAST".*

p.36,l.754 - I suspect the reference needs to moved to '. . . new nutrients (e.g. . . . ) and ' - since the study of Moutin took place before your studies

*Corrected*

p.37,l.777 - more important → higher (or, in which sense more important?)

*Corrected*

p.38,l.790 - why don't you refer to the decreasing TEP-C as shown in Fig. 3?

*Indeed, this would be better. Corrected.*

p.38,l.799-804 - The reference in line 801 cannot really refer to your experiments, so I would suggest to reformulate this part in terms of what Bressac and Guieu 2013 found and how that relates to your study.

*Agree. The sentence was reformulated (see also comment above), a different reference was given:*
*(sentence L796-801) was replaced by:*
*"The recovery of the introduced dust (traced by the lithogenic mass recovered in the traps) was low (~30% at TYR, ~20% at ION and 15-19% at FAST) reflecting that a majority of the dust particles (the smaller ones that are the most abundant according to the particle size distribution of the dust) still remained in the tanks after 3 or 4 days following dust addition. This has been already observed in pelagic mesocosms (Bressac et al., 2012) as those small particles can aggregate to organic matter and eventually sink. The higher export efficiency observed (TYR>ION>FAST) is likely linked to the higher initial abundance and higher production of TEPs during the experiment (Fig. 3).*

*Ref: Bressac, M., Guieu, C., Doxaran, D., Bourrin, F., Obolensky, G., & Grisoni, J. M. (2012). A mesocosm experiment coupled with optical measurements to assess the fate and sinking of atmospheric particles in clear oligotrophic waters. Geo-Marine Letters, 32(2), 153-164.*

p.39,l.822 - this treatment → future conditions (or treatment G compared to . . . )

*Corrected: "Similarly, the heterotrophic compartment was more stimulated, as BP rates increased strongly at all stations under future conditions compared to treatment D. "*

p.42,l.899 -  weakening of the CO2. .

*Corrected.*

Specific comments on figures and tables

Fig.4 I cannot distinguish between the individual tanks. I suspect the order matters. Since you refer to individual tanks in the text, it would be helpful to be able to distinguish between the tanks.

*Figures 4, 6 and 8 have been updated in order to distinguish the two replicates (empty vs full bars). It was less problematic for Fig. 8 (as the names of the tanks are on the x axis, but it is important to be homogenous).*

.

---

## Author Response (AR2)

Dear editors, many thanks for your comments and corrections on our manuscript. Please see below the answers to your comments and the associated changes made to our manuscript.

Kind regards
FG

- At line 353, the reference is missing.

*After checking, this seems to be the line 309. Reference added: "Schimmelmann, A., Qi, H., Coplen, T. B., Brand, W. A., Fong, J., Meier-Augenstein, W., Kemp, H. F., Toman, B., Ackermann, A., Assonov, S., Aerts-Bijma, A. T., Brejcha, R., Chikaraishi, Y., Darwish, T., Elsner, M., Gehre, M., Geilmann, H., Gröning, M., Hélie, J.-F., Herrero-Martín, S., Meijer, H. A. J., Sauer, P. E., Sessions, A. L., and Werner, R. A.: Organic Reference Materials for Hydrogen, Carbon, and Nitrogen Stable Isotope-Ratio Measurements: Caffeines, n-Alkanes, Fatty Acid Methyl Esters, Glycines, L-Valines, Polyethylenes, and Oils, Anal. Chem., 88, 4294–4302, https://doi.org/10.1021/acs.analchem.5b04392, 2016."*

- In the reference list please complete the reference Desboeufs, 2022.

*Reference updated: "Desboeufs, K.: Nutrients atmospheric deposition and variability, in Atmospheric Chemistry in the Mediterranean – Vol. 2, From Pollutant Sources to Impacts, edited by Dulac, F., Sauvage, S., and Hamonou, E., Springer, Cham, Switzerland, in press, 2021."*

- Two reviewers have the same issue with your expression for GPP=NCP-CR, in particular because you previously define "The metabolic balance (or net community production, NCP) is defined as the difference between gross primary production (GPP) of autotrophic organisms and community respiration (CR) of both autotrophic and heterotrophic organisms, revealing the capacity of surface waters to absorb atmospheric CO2." The description in the method approach leads to confusion. I recommend that after "NCP and CR were estimated by regressing O2 values against time." you add the following sentence "Since CR is estimated from the oxygen evolution (consumption) in bottles (negative sign), GPP corresponds to, and was calculated as GPP= NCP-CR "

*Added to the text.*

- For the sediment trap data processing please explain how "the samples where rinsed to remove sea salt" (centrifugation/resuspension in freshwater/MilliQ?) or give a reference.

*Added to the text: "the samples were rinsed by centrifugation in MilliQ water (3 times) to remove sea salt"*

- Line 309: A reference is missing

*Reference added: "Schimmelmann, A., Qi, H., Coplen, T. B., Brand, W. A., Fong, J., Meier-Augenstein, W., Kemp, H. F., Toman, B., Ackermann, A., Assonov, S., Aerts-Bijma, A. T., Brejcha, R., Chikaraishi, Y., Darwish, T., Elsner, M., Gehre, M., Geilmann, H., Gröning, M., Hélie, J.-F., Herrero-Martín, S., Meijer, H. A. J., Sauer, P. E., Sessions, A. L., and Werner, R.*

*A.: Organic Reference Materials for Hydrogen, Carbon, and Nitrogen Stable Isotope-Ratio Measurements: Caffeines, n-Alkanes, Fatty Acid Methyl Esters, Glycines, L-Valines, Polyethylenes, and Oils, Anal. Chem., 88, 4294–4302, https://doi.org/10.1021/acs.analchem.5b04392, 2016."*

-Please provide the information given in your answer to reviewer #3 in the text: "L345 to 356: please quantify the blanks, agreement between replicates and the standard deviation The blanks were 1.1 % of the average concentration of the sample (thus negligible), replicates agreement was on average 0.3 - 2.3 % (and no standard deviation as we analyse only 2 aliquots over the 3 when the 2 first measurements agree (<5% difference) and that was the case for all the samples analysed in this study. "

*Added at the end of the section.*

-Please (as requested by reviewer #4) add the date of the cruise in the figure 1 or in the figure legend.

*Added to the figure legend.*